# A live bacteria enzyme assay for identification of human disease mutations and drug screening

Donghui Choe [1] & Bernhard O. Palsson [1,2] ✉

Advances in high-throughput sequencing have enabled the identification of genetic variations associated with human disease. However, deciphering the functional significance of these variations remains challenging. Here we propose an alternative approach that uses humanized *Escherichia coli* to study human genetic enzymopathies and to screen candidate drug effects on metabolic targets. By replacing selected *E. coli* metabolic enzymes with their human orthologues and their sequence variants, we demonstrate that the growth rate of *E. coli* reflects the in vivo activity of heterologously expressed human enzymes. This approach accurately reflected enzyme activities of known sequence variants, enabling rapid screening of causal sequence variations associated with human diseases. This approach bridges the gap between in vitro assays and cell-based assays. Our findings suggest that the proposed approach using a humanized *E. coli* strain holds promise for drug discovery, offering a high-throughput and cost-effective platform for identifying new compounds targeting human enzymes. Continued research and innovation in this field have the potential to impact the development and practice of precision medicine.

The human genome sequence varies between individuals. Individuals typically harbour between 250 and 300 loss-of-function sequence variations relative to the reference human genome[1]. These variations often underlie inherited disorders, making the identification of causal mutations an essential pursuit in human biology and personalized medicine. Recent strides in genome editing offer promising avenues for treating conditions such as eye and liver diseases[2,3].

Traditionally, the identification of genotypes associated with genetic disease has relied on direct patient analysis and case reports. Once genetic variations linked to specific disorders are mapped, they are studied through various methodologies. One common approach involves in vitro studies with patient-derived materials or recombinant proteins to characterize mutant proteins. However, this approach is constrained by sample availability and necessitates labour-intensive purification processes. Moreover, it often requires specialized assay systems tailored to individual proteins, alongside time-consuming analytical methods and instrumentation. In addition, dilute in vitro conditions do not replicate crowded and dense gel-like physiological conditions, necessitating an in vivo system for an accurate biochemical assay[4].

Alternatively, large-scale computational surveys, such as genome-wide association studies based on population sequencing datasets[5], effectively identify potential associations between genetic changes and pathological conditions[6]. However, such associations do not establish causality, particularly given the many uncharacterized variations and confounding factors of zygosity and genetic linkage[5,7]. Furthermore, variations that induce severe pathogenicity from early onset are rare in the human population, making it challenging to obtain samples or detect such variations in population databases. As a result, pathogenic mutations remain practically undetectable, while detected mutations are more likely to have minor impact on enzyme activity.

[1]Shu Chien-Gene Lay Department of Bioengineering, University of California San Diego, La Jolla, CA, USA. [2]Department of Pediatrics, University of California San Diego, La Jolla, CA, USA. ✉e-mail: bpalsson@ucsd.edu

In prior work, we demonstrated the replacement of *Escherichia coli* glycolytic genes with their human orthologues[8]. Despite the evolutionary distance between bacteria and humans, they share substantial metabolic similarities, including a universal glycolytic framework across phyla, where glucose is metabolized through identical chemical reactions, albeit with different enzymes. Thus, *E. coli* lacking key native enzymes for glucose metabolism could use glucose with the introduction of corresponding human enzymes without alterations to their coding sequences.

Thus, we proposed an alternative approach to investigate the activities of human mutant enzymes by developing a live *E. coli* assay (LEICA). Specifically, we focused on glucose-6-phosphate isomerase (GPI) and glucose-6-phosphate dehydrogenase (G6PD), which are associated with the most common human hereditary enzymopathies[9,10].

As the growth rate of *E. coli* is contingent upon glycolytic flux[11], it serves as a surrogate measure for the activity of heterologously expressed human enzymes. Harnessing the ease of genetic manipulation in *E. coli*, we replicated human mutations in these enzymes. *E. coli* strains carrying different mutants exhibited distinct growth rates, reflecting variations in enzyme activity induced by mutations. Notably, the growth rates demonstrated a high linear correlation with enzyme activities determined via recombinant protein assays. This live bacterial cell assay provides an accurate and rapid means of screening for enzyme activity change resulting from human mutations, offering insights into the causality of genetic disorders attributable to genetic variations. We also expanded LEICA to screen argininosuccinate lyase (ASL), a key urea cycle enzyme whose deficiency results in argininosuccinic aciduria[12]. Complementation of arginine auxotrophy in *E. coli* lacking *ASL* with human equivalents demonstrates LEICA's broader applicability across diverse enzymes.

LEICA revealed other potential uses. Through experimentation with small molecules using LEICA, we found that chemical compounds targeting human G6PD could either enhance or inhibit growth when administered directly to cells in culture. We confirmed inhibitory effects of known G6PD inhibitors, highlighting the drug screening capability of LEICA on human drug targets. Screening a library of 160 human metabolism modulators revealed 7 lead compounds, including the rediscovery of 3 with known G6PD-inhibitory effects or antimalarial activity. Lastly, we observed enhanced growth of *E. coli* carrying the G6PD-Canton mutant when treated with the recently discovered G6PD agonist AG1 (ref. 13). LEICA thus not only serves as a drug screening platform but also holds promise for use in developing personalized medicine.

## Results

**Growth of *E. coli* with human GPI reflects enzyme activity.** In a previous study, we demonstrated the successful replacement of *E. coli pgi*, which encodes for GPI, with its human orthologue[8]. During adaptive laboratory evolution, optimal human GPI function in live bacteria required enhanced gene expression rather than mutations in the protein sequence. Furthermore, various in vitro studies use recombinant human enzymes expressed from *E. coli* lysates[14,15], indicating *E. coli* as a potential host for functional expression of human enzymes.

In both humans and *E. coli*, glucose is mainly utilized by glycolysis. Thus, we hypothesized that replacing a glycolytic enzyme in *E. coli* with its human counterpart would reflect human enzyme activity (Fig. 1a). To specifically assess human GPI activity, we disengage the hexose monophosphate (HMP) shunt by deleting *zwf*, encoding G6PD. We thus knocked out *zwf* from the *E. coli* strain 20.71—an evolved *E. coli* K-12 MG1655 strain[16] carrying in-frame *pgi* swap to human *GPI* and promoter mutations (Fig. 1b). In the resulting strain—a humanized *E. coli* for GPI—growth in a glucose-containing medium is solely dependent on GPI activity (Extended Data Fig. 1).

We examined genetic variations in *GPI* associated with haemolytic anaemia, selecting six mutations including two benign and four

pathogenic ones (Supplementary Table 1)[14]. Two benign mutations occurred with population frequencies ranging from 1 in 49 to 1 in 4,400 individuals, while 4 pathogenic variations ranged from 1 in 57,000 to 1 in 390,000 (Genome Aggregation Database, gnomAD[17]). Some were reported in patients with haemolysis[18–20].

To evaluate the live bacterial assay, we replaced wild-type (WT) human *GPI* in the humanized *E. coli* for GPI with the mutant *GPI*s to investigate activity differences. Strains expressing different GPI variants showed growth rate changes ranging from −13.1% to +2.2% compared with the WT control (Fig. 1c). Growth rates of strains carrying pathogenic mutations were significantly lower than those with WT GPI, while two benign mutants displayed either a mild decrease in growth (−5.6%) or growth indistinguishable from the WT control.

Comparison of biochemically determined properties of recombinant mutants[14] with the live bacterial assay revealed a high linear correlation between enzyme activity and the growth rate, demonstrating the assay's capability to infer human enzyme activities (Fig. 1d). The LEICA gives an impetus for rapidly screening genetic variants causing enzyme deficiencies. Unlike conventional biochemical assays using red blood cell (RBC) haemolysates or recombinant enzymes that require specialized systems and laborious purification[21], LEICA calls for only a single gene replacement (and variants therein) with a straightforward growth measurement, as all the necessary substrates and cofactors are present in live bacteria.

Having demonstrated the concept of LEICA analysing human enzyme activity, we sought to explore another human enzyme, G6PD. G6PD deficiency is the most common cause of RBC enzymopathy, affecting 300–400 million people worldwide[22]. G6PD plays a critical metabolic role in RBCs as the HMP shunt provides NADPH needed for redox homeostasis[23]. Hereditary G6PD deficiency is caused by sequence variations in *G6PD*, which is highly polymorphic with over 230 variants identified[6,10], due to its association with malaria resistance[24]. Despite many variants identified through populational sequencing, clinical or molecular characterizations remain limited[6].

To assess *G6PD* variations using LEICA, we first examined whether human G6PD could functionally express in *E. coli* (Fig. 1e). As the *E. coli* 20.71 Δ*pgi* Δ*zwf* strain is unable to utilize glucose as a carbon source (Extended Data Fig. 1), growth in a glucose medium directly reflects heterologous G6PD activity, creating a humanized *E. coli* for G6PD. The double-knockout strain expressing human G6PD exhibited comparable growth to the *E. coli* Δ*pgi* single-knockout strain, whose glycolytic flux is supported by its endogenous G6PD (Zwf) (Extended Data Fig. 2). As demonstrated with the *GPI* gene replacement, human G6PD proved functional in *E. coli*.

Next, we selected 13 human *G6PD* sequence variations associated with varying severities of G6PD deficiency for LEICA application (Supplementary Table 2)[6]. Strains expressing G6PD variants showed markedly different growth profiles to those carrying the WT enzyme (Extended Data Fig. 3a,b). Specifically, the humanized *E. coli* carrying G6PD-Volendam, a class I G6PD variant associated with chronic haemolysis, grew at half the rate of the WT-expressing strain (Fig. 1f). Growth of the relatively mild deficiency variant, G6PD-Mahidol (class III) was also apparent (84% of WT, $P = 1.412 \times 10^{-4}$, Welch's *t*-test), indicating the high dynamic range of the growth assay. Moreover, growth rates of humanized *E. coli* carrying different G6PD variants correlated linearly with biochemical properties of the corresponding recombinant enzymes (Pearson's $R^2$ of 0.84; Fig. 1f)[25–33]. This result further illustrates LEICA's ability to rapidly screen human genetic variants.

Given the linear correlation observed, we sought to test less-characterized variants with no reported biochemical properties using LEICA. We constructed humanized *E. coli* strains carrying two predicted benign variants, one pathogenic variant and one variant with uncertain clinical significance (Supplementary Table 2). LEICA detected marginal growth rate reductions in the two predicted benign variants (Fig. 1g and Extended Data Fig. 3b), indicating a mild decrease

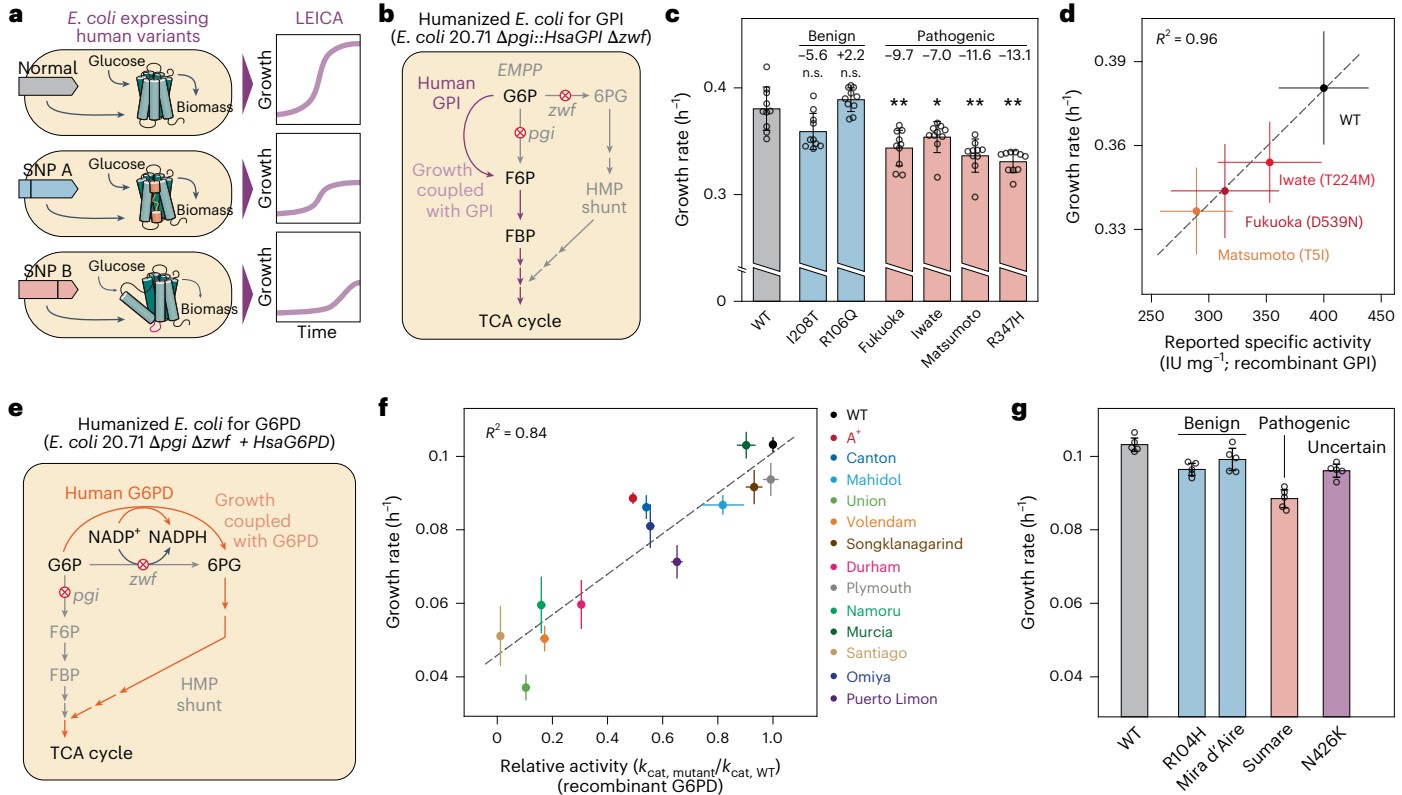

**Fig. 1 | A bacterial system represents activities of human enzymes by growth rate. a**, The bacterial system used for the screening of activities of genetic variants of human metabolic enzymes. SNP, single-nucleotide polymorphism. **b**, Growth of the *E. coli* phosphoglucose isomerase (encoded by *pgi*) and G6PD (encoded by *zwf*) double-knockout strain is dependent on human GPI activity. EMPP, the Embden–Meyerhof–Parnas pathway; G6P, glucose-6-phosphate; 6PG: 6-phosphogluconate; F6P, fructose-6-phosphate; FBP, fructose-1,6-bisphosphate; TCA, tricarboxylic acid. **c**, Activities of human GPI variants were represented by growth rates of the humanized *E. coli* for GPI. Data are presented as mean values ± s.d. Error bars indicate the s.d. of ten replicated cultures. Numbers are relative differences of growth rates compared with the WT enzyme; in percentage. n.s., no significant difference. *$P = 0.022$; **$P = 0.002$ (Fukuoka), 0.000 (Matsumoto) and 0.000 (R347H) (two-sided Welch's *t*-test with Bonferroni correction). Circles indicate ten independent cultures. **d**, Growth rates of the

*E. coli* with human gene swap show high linear correlation with previously reported activities of recombinant enzymes (Pearson's $R^2$ of 0.96; dashed line)[14]. Data are presented as mean values ± s.d. Error bars indicate the s.d. of biological replicates ($n = 10$). Specific activities of recombinant enzymes were pulled from a previous report[14]. **e**, Growth of the double-knockout strain expressing human G6PD is governed by activity of the human G6PD. **f**, Growth rates of humanized *E. coli* for G6PD and catalytic constants ($k_{cat}$) of recombinant enzymes[25–30] had high correlation (Pearson's $R^2$ of 0.84; dashed linear regression line). Data are presented as mean values ± s.d. Error bars indicate the s.d. of five replicated cultures. For G6PD-Volendam, the specific activity was used[31], because $k_{cat}$ is not available. **g**, Activities of a few less-characterized human G6PD variants were represented by growth rates of the humanized *E. coli*. Data are presented as mean values ± s.d. Error bars indicate the s.d. of five replicated cultures. Circles indicate independent data points.

---

in enzyme activity. However, the association of these variants with disease should be considered carefully, as LEICA primarily screens for enzyme activity. If these mutations are revealed to be clinically associated with diseases, it may imply that pathogenicity arises from factors other than enzyme activity decrease, such as enzyme stability or expression level. In addition, a variant with insufficient evidence to determine its role in disease (N426K) displayed similar results to the benign variants, suggesting that N426K induces no significant enzyme deficiency.

By contrast, the G6PD-Sumare variant exhibited a notable reduction in growth (86% of WT, $P = 6.736 \times 10^{-5}$, Welch's *t*-test), comparable to that of class III variants (A⁺, Mahidol and Murcia) associated with mild anaemia[34]. Although the biochemical properties have never been studied, the results from LEICA, along with reduced blood G6PD activity in patients carrying this variation, indicate a possible association of the G6PD-Sumare variant with disease[35].

Incorporating G6PD and GPI into LEICA enables accurate measurement of catalytic activities of human enzymes in an intracellular environment. Unlike an in vitro assay, the bacterial host provides the necessary substrates and physiological conditions for enzyme activity. This approach allows for an in vivo enzyme assay by simple growth

measurement, independent of specialized chemical and reporter systems. Human genetic disorders result from a complex interplay of biological factors, including variations in catalytic activity, expression levels and zygosity, posing challenges in identifying causality. LEICA insulates enzyme activity from such biological factors by mimicking the homozygous state and providing stable expression levels, offering a rapid means of assessing only catalytic activity as the primary variable.

Human G6PD is functional only as a dimer or tetramer, and some mutations hamper oligomerization[36]. A small-molecule activator of G6PD, AG1, promotes dimer formation and activates mutants with reduced oligomerization, such as G6PD-Canton[13,15]. AG1 improved G6PD-Canton activity up to 1.7-fold with half-maximal effective concentration (EC₅₀) of 3.4 μM and increased WT G6PD activity by approximately 20% in previous study[15]. Thus, we examined the effect of AG1 using LEICA to assess its suitability for screening small molecules.

Exposure to AG1 had no effect on the growth of *E. coli* with endogenous *G6PD* (*E. coli* 20.71 Δ*pgi*) ruling out its impact on bacterial metabolism (Fig. 2). In humanized *E. coli*, AG1 improved the growth rates of strains expressing WT G6PD and Canton variant up to 12.8% and 20.1% at 0.3 μM, respectively (Fig. 2a), a result that is consistent with the previous study (Supplementary Note 1)[15]. Activated G6PD-Canton supported

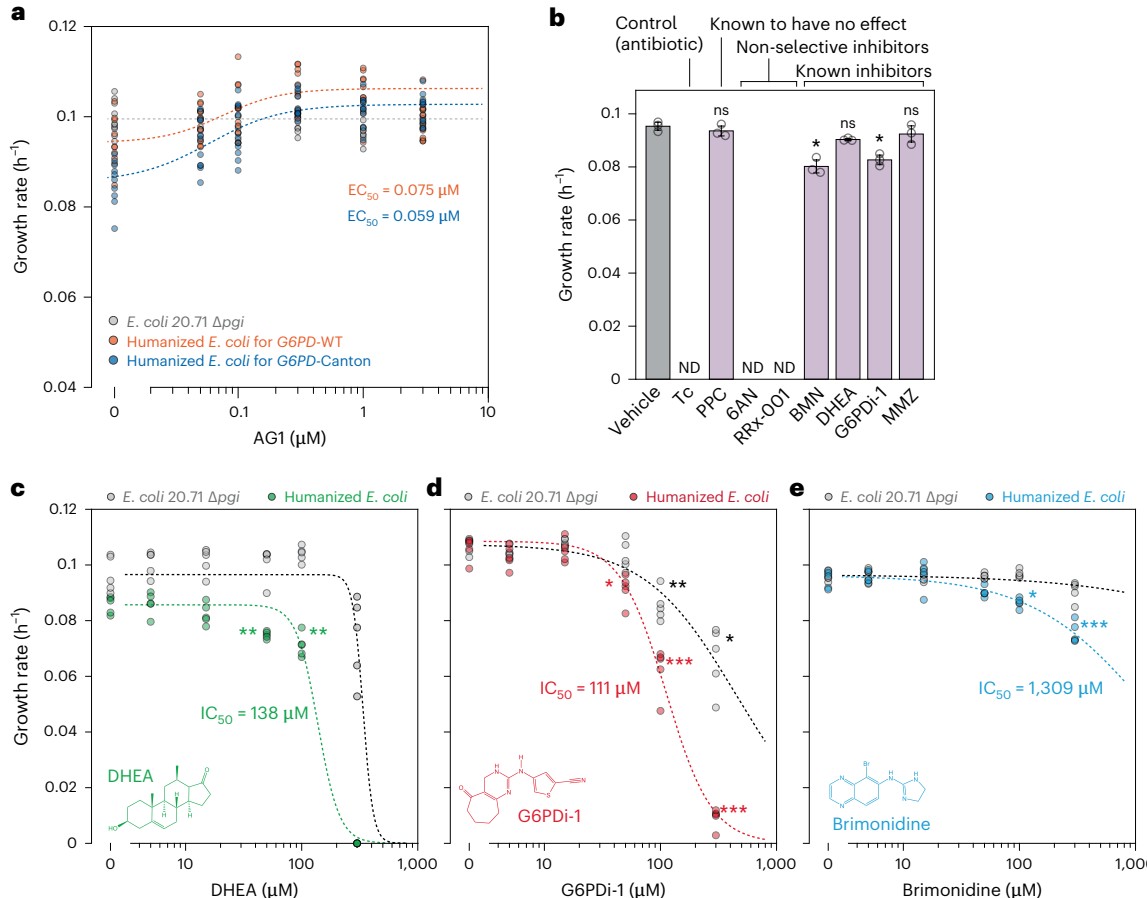

**Fig. 2 | Effect of small molecules on G6PD. a**, The effect of a G6PD activator AG1 on the growth rate of humanized *E. coli*. $EC_{50}$ is calculated from the dose–response curve (dashed lines). Circles are individual data points ($n = 10$). **b**, Growth rates of humanized *E. coli* in LEICA with different compounds. Vehicle: untreated control (1% DMSO). All compounds were treated with the final concentration of 100 μM, except for Tc (10 μg ml⁻¹). ND, no observable growth was detected. Data are presented as mean values ± s.d. Error bars show the s.d. of three replicates. Dots are individual data points. n.s., no significant difference.

*$P = 0.025$ (BMN) and 0.0135 (G6PDi-1) (two-sided Welch's *t*-test with Bonferroni correction, compared with the untreated control). **c–e**, Dose–response validation of the inhibitory effect of three primary hits G6PDi-1 (**c**), DHEA (**d**) and BMN (**e**). $IC_{50}$ was calculated from a Hill curve (four parameters; dashed line) fitted to the dose–response curve. Dots are individual data points ($n = 5$). *$P < 0.05$, **$P < 0.005$, ***$P < 0.001$ (two-sided Welch's *t*-test with Bonferroni correction, compared with the untreated control).

the growth of humanized *E. coli* better than untreated WT, and there was no significant difference in the activities when maximally activated ($P = 0.055$, Welch's *t*-test; Fig. 2a). Notably, LEICA required lower AG1 concentrations than in vitro and cell-based screens[15], probably due to differences in reaction constituents concentrations (for example, substrates, cofactors and enzymes) or compound bioavailability across bacterial and mammalian membranes. Nonetheless, LEICA successfully reflected G6PD activation by AG1 without the need for specialized reporter-based assay system.

Given that LEICA can measure the effect of AG1 on human G6PD, we aimed to evaluate its ability to screen various compounds that target G6PD. As G6PD-deficient patients face a high risk of haemolysis when exposed to certain drugs, specific foods or infections[37], it is important to evaluate the impact of drugs on G6PD when devising treatment strategies for G6PD-deficient individuals. In addition, drugs that act on G6PD are active areas of clinical research, as it is linked to antimalarials, immune response and tumours[38–41].

Seven compounds with known effects on G6PD were chosen for screening (Supplementary Note 2). These include: (1) the local anaesthetic proparacaine (PPC), (2) the non-selective inhibitor of NADP-dependent enzymes 6-aminonicotinamide (6AN), (3) the energetic compound, radiosensitizer bromonitrozidine (RRx-001), (4) the $\alpha_2$-adrenergic agonist brimonidine (BMN), (5) the steroid hormone

dehydroepiandrosterone (DHEA), (6) the small molecule G6PD inhibitor of G6PD 1 (G6PDi-1) and (7) the pyrazolonic analgesic metamizol (MMZ). Previous reports indicate that PPC has no significant effect on G6PD[42], while G6PDi-1[41], DHEA[43], BMN[42] and MMZ[44] exhibit inhibitory effects. RRx-001 and 6AN are non-selective inhibitors[45,46], either generating reactive radicals or non-selectively inhibiting NADP-dependent enzymes. Each compound was supplemented to culture media at a concentration of 100 μM and 10 μg ml⁻¹ tetracycline (Tc) was used as a positive control for bacterial growth inhibition.

LEICA treated with PPC exhibited no difference in growth profile compared with the untreated control (Fig. 2b and Extended Data Fig. 4), aligning with previous observations[42]. Treatment with Tc, RRx-001 and 6AN completely halted the growth of humanized *E. coli* for G6PD in LEICA (Fig. 2b). However, the growth of *E. coli* Δ*pgi* was also suppressed by RRx-001 and 6AN (Extended Data Fig. 5), indicating their toxicity to *E. coli* and prompting us to exclude them as hits.

The growth of humanized *E. coli* for G6PD (WT) was significantly inhibited by G6PDi-1, DHEA and BMN (Fig. 2b,c and Extended Data Fig. 4), with no effect observed on *E. coli* Δ*pgi* (Extended Data Fig. 5), underscoring the inhibitory effect of these compounds on human G6PD, consistent with the previous reports[41–43]. However, MMZ showed no discernible effect. This is probably attributable to the high inhibitory concentration of MMZ (half-maximal inhibitory concentration ($IC_{50}$)

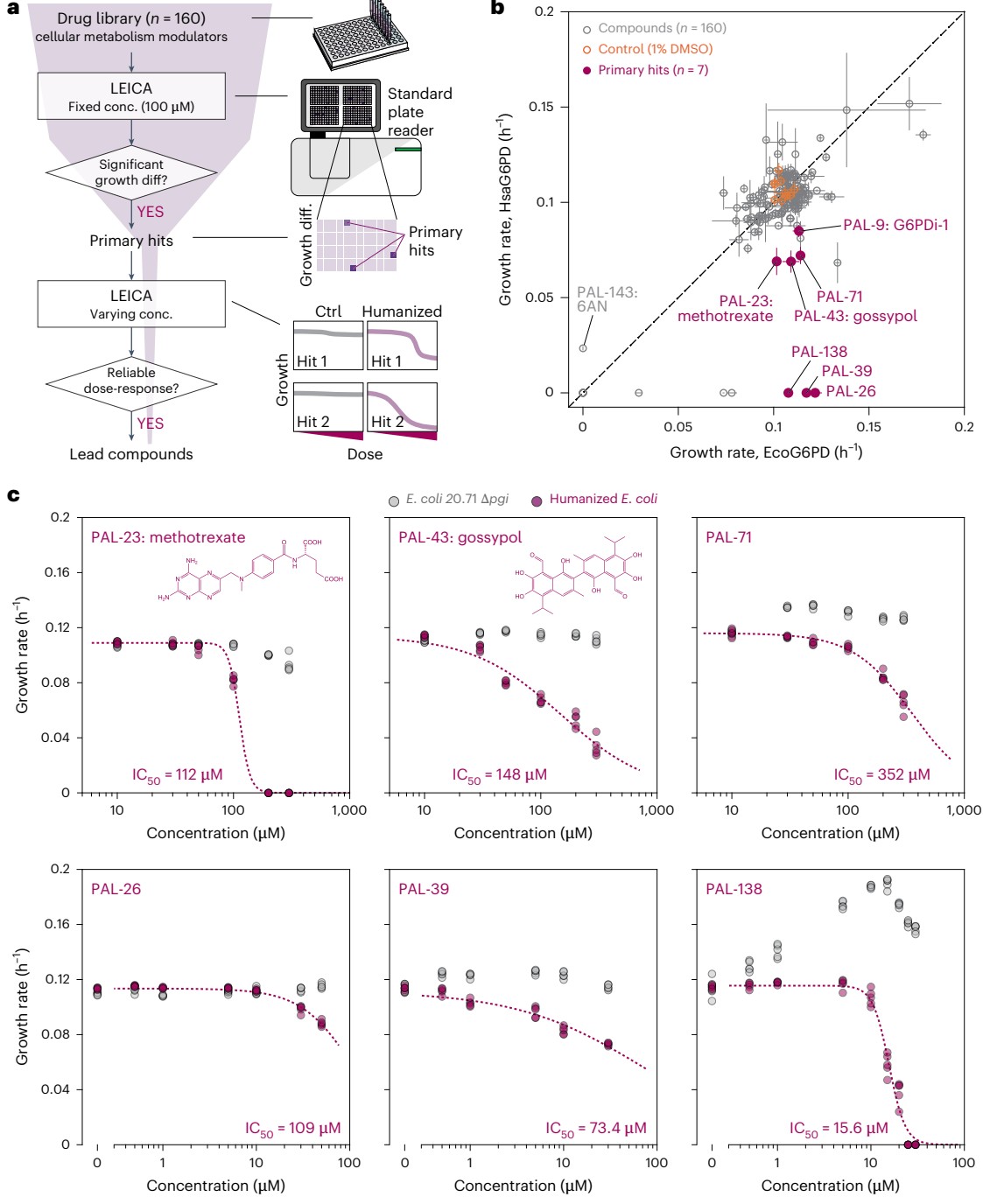

**Fig. 3 | Drug library screening using LEICA. a**, Various compounds are supplemented to LEICA, and growth is monitored using a plate reader. The change in G6PD activity in response to chemical compounds is reported as the growth difference (diff) of the humanized *E. coli* at a fixed concentration compared with the untreated control (primary hits). Primary hits are further validated by dose–response assay, resulting in discovery of lead compounds. **b**, Growth rates of *E. coli* expressing human G6PD compared with that of *E. coli* expressing endogenous G6PD. Seven compounds induced significant ($P < 0.001$; two-sided Welch's *t*-test, compared with the untreated control) growth difference, while inducing no significant difference on *E. coli* expressing endogenous G6PD. All compounds were treated with the final concentration of 100 μM. Data are presented as mean values ± s.d. Error bars show the s.d. of three replicates. **c**, The effect of primary hit compounds at different concentrations on humanized *E. coli* for G6PD WT. $IC_{50}$ is calculated from the dose–response curve (dashed lines). Circles are individual data points ($n = 5$).

of 17 mM) required to inhibit G6PD activity[44], in contrast to G6PDi-1 and BMN, whose $IC_{50}$s are in the submillimolar range (0.07–30 μM).

Two compounds—G6PDi-1 and BMN—which exhibited inhibitory effects in the primary screen, underwent further examination in dose–response assays to validate the screening results. We also included DHEA, which has been reported to have varying G6PD-inhibitory

effects depending on the assay format. LEICA was performed with each compound at concentrations ranging from 5 to 300 μM, revealing a dose-dependent response with $IC_{50}$ of 111, 138 and 1309 μM, respectively (Fig. 2c–e). These $IC_{50}$ values differed from those obtained in in vitro assays where the $IC_{50}$s of G6PDi-1, DHEA, and BMN were reported as 0.07, 9 and 30 μM, respectively[41,42]. However, $IC_{50}$ values measured

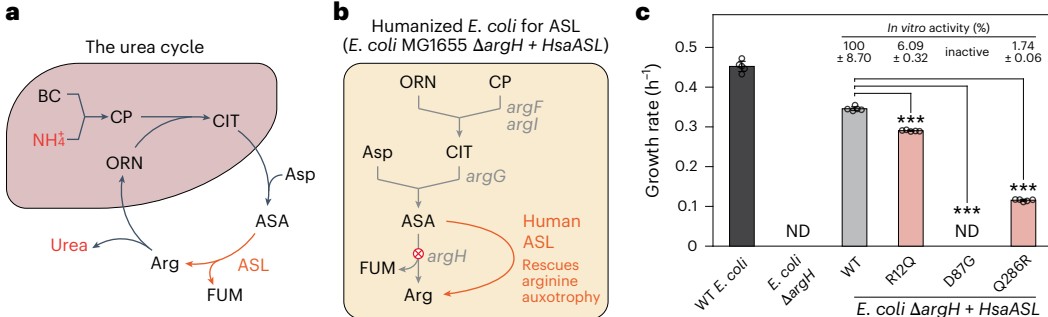

**Fig. 4 | Expanding LEICA for screening human ASL. a**, A schematic representation of the human urea cycle. BC, bicarbonate; CP, carbamoyl phosphate; CIT, citrulline; ASA, argininosuccinate; ORN, ornithine; Fum, fumarate. **b**, Arginine auxotrophy induced by *argH* knockout is complemented by human ASL. **c**, Growth rates of WT *E. coli*, *argH*-knockout strain and *argH*-knockout strain expressing WT and variants of human ASL. In vitro activities of the recombinant enzymes[12,52] are indicated above. ND, no observable growth was detected. Data are presented as mean values ± s.d. Error bars show the s.d. of five replicates. Dots are individual data points. ***$P$ = 0.000 (two-sided Welch's $t$-test with Bonferroni correction, compared with the WT).

from LEICA are more closely aligned with results from cell-based assays, where the IC$_{50}$ value for G6PDi-1 ranged from 13 to 31 µM depending on the cell type. Interestingly, DHEA exhibited an inhibitory effect in LEICA, contrary to the lack of effect in cell-based assays[41]. Previous reports indicate that the antiproliferative effects of DHEA on various cells may stem from indirect effects that are not fully understood[47]. Thus, the behaviour of LEICA in this context resembles that of a recombinant assay. Despite these discrepancies, LEICA effectively screened compounds with comparable IC$_{50}$ values, making it a suitable format for high-throughput screening applications.

Having demonstrated the sensitivity of LEICA to known compounds that modulate G6PD activity, we screened a drug compound library to identify lead compounds with potential effects on G6PD. The screening involved two stages: (1) a single-dose growth assay using LEICA, and (2) validation of primary hits through dose–response assays (Fig. 3a). Among the 160 human metabolism modulators screened, 7 compounds (primary hits) were identified that inhibited the growth of *E. coli* expressing human *G6PD*, while showing no effect on the strain with endogenous *E. coli G6PD* (Fig. 3b).

The screening successfully rediscovered G6PDi-1 as a primary hit, confirming the assay's robustness. Two other primary hits, methotrexate and gossypol, have also been previously linked to G6PD. Methotrexate, a dihydrofolate reductase inhibitor, has reported G6PD-inhibitory effects (IC$_{50}$ of 114 µM)[48] and antimalarial activity[49]. Gossypol, known for its antimalarial properties, inhibits oxidoreductases, including G6PD in *Trypanosoma cruzi*[50]. Given that G6PDi-1 was originally discovered as an inhibitor of *T. cruzi* G6PD[51], it logically follows that gossypol, with its reported effects on *T. cruzi*, might also inhibit human G6PD. These three previously known compounds detected in the screen validate the sensitivity and reliability of the assay.

The four remaining compounds have no previously reported association with G6PD activity; however, three completely inhibited the growth of *E. coli* expressing human G6PD. Further characterization using dose–response assays revealed reliable dose-dependent effects from all the primary hits (Fig. 3c). Among the four identified compounds, three displayed higher potency with IC$_{50}$s ranging from 15.6 to 109 µM, compared with G6PDi-1, methotrexate and gossypol (IC$_{50}$s of 111–148 µM). This result suggests that these identified compounds are candidate leads for antimalarial drugs as potent G6PD inhibitors. The drug screening demonstrates LEICA's potential as a versatile high-throughput screening tool capable of identifying compounds that influence human enzymes within a bacterial context.

To broaden LEICA's applicability to enzymes outside glycolysis, we focused on ASL, a critical enzyme in the urea cycle (Fig. 4a)[52]. Although *E. coli* does not have a urea cycle, it contains an ASL enzyme encoded by the *argH* gene, which is involved in arginine biosynthesis. As knockout

of *argH* results in arginine auxotrophy, it creates a suitable platform for assaying human ASL activity by growing cells in arginine-free conditions (Fig. 4b). We found that human ASL could complement the arginine auxotrophy in an *E. coli-argH*-knockout strain, indicating successful functional expression of human ASL in *E. coli* (Extended Data Fig. 6).

We then examined various ASL sequence variants that are known to cause argininosuccinic aciduria, with residual enzyme activity ranging from 0% to 6.09% (Supplementary Table 3)[12,52]. The humanized *E. coli* for ASL whose arginine biosynthesis is supported by different pathogenic ASL variants exhibited significant growth reductions (ranging from 16% to 100%) compared with cells expressing WT human ASL (Fig. 4c and Extended Data Fig. 6). Specifically, the growth rates of humanized *E. coli* aligned with the residual activities of recombinant ASL variants: D87G, which results in a complete loss of activity, could not complement the arginine auxotrophy, while the least severe R12Q variant supported growth to 84% of WT levels.

The ASL example demonstrates LEICA's versatility in screening enzymes from diverse metabolic pathways. Its expandability suggests that LEICA could be used as a platform to study a wide range of human metabolic enzymes, further broadening the potential applications in identifying pathogenic sequence variations and therapeutic compounds for metabolic disorders.

## Discussion

High-throughput sequencing has revolutionized our understanding of the human genome, providing vast amounts of data on genetic variations. However, the key challenge lies in deciphering the functional importance of these variations, particularly those associated with genetic disorders. Genome-wide association studies have identified mutations linked to various disorders, but understanding the causality of these mutations remains a substantial hurdle.

To address this, we developed LEICA, a live bacteria assay to measure activities of heterologously expressed human enzymes. Unlike traditional in vitro assays, LEICA leverages bacterial metabolism to rapidly characterize genetic variations associated with enzyme deficiencies. By coupling bacterial growth with enzyme activity, LEICA provides a simpler, cost-effective alternative that aligns well with prior reports. Its ability to screen pathogenic variations through this coupling makes it a distinct improvement over traditional methods, by rapidly providing insights into the functional consequences of genetic variations.

Beyond its use in studying genetic variations, LEICA proves effective for screening drug effects on human enzymes. By directly assessing the impact of various compounds on enzyme activity in an intracellular environment, LEICA bridges the gap between in vitro and cell-based assays. While offering advantages such as simplicity, speed

and cost-effectiveness, it also presents some limitations, such as its inability to screen compounds with antimicrobial properties. However, incorporating pairwise comparisons between humanized *E. coli* and the strain carrying the endogenous bacterial gene effectively prevents false positives, as demonstrated with 6AN treatment (Figs. 2b and 3b and Extended Data Fig. 5). Also, we acknowledge that bacterial cells may have different permeability to chemical compounds compared with human cells, which could affect the assay's representation of drug efficacy in human cells. However, *E. coli* cells are non-selectively permeable to small molecules (<600 Da)[53], and LEICA's demonstrated robustness in identifying seven effective compounds from the drug library supports its reliability as a screening tool. Nonetheless, LEICA's ability to operate under intracellular conditions makes it a valuable approach for compound screening. Looking ahead, LEICA holds notable promise for high-throughput screening and personalized drug testing (Supplementary Note 3).

In conclusion, LEICA represents an alternative approach for studying genetic enzymopathies, screening drug effects and advancing precision medicine. By leveraging bacterial metabolic machinery, LEICA offers insights into the functional consequences of genetic variations and facilitates the rapid screening of drug libraries. Moving forward, continued research and innovation in this field holds the potential to impact the identification of lead compounds for human targets that can be manifested in *E. coli*.

## Methods

### Bacterial strains and culture conditions
*E. coli* strain 20.71, which is an adaptively evolved strain with an in-frame substitution of its endogenous phosphoglucose isomerase with human homologue, was constructed in a previous study[8]. The *E. coli* 20.71 Δ*pgi* strain was constructed by substituting the human *GPI* with the *sacB-cat* dual-selection cassette using lambda recombination[54]. The *E. coli* 20.71 Δ*pgi* Δzwf double-knockout strain was constructed by substituting the *zwf* gene with a kanamycin-resistance cassette on the *E. coli* 20.71 Δ*pgi* background using lambda recombination. Humanized *E. coli* carrying either the WT GPI or GPI variants were constructed by introducing human WT *GPI* or variant genes into the double-knockout strain using lambda recombination. Human *GPI* variants were constructed by assembling split human *GPI* fragments (upstream and downstream of the variation) amplified by primers containing genetic variations. Full *GPI* constructs were assembled using overlap extension PCR. In brief, 5 ng each of the fragments were first ligated by 15 cycles of the following PCR reaction: 98 °C for 30 s, 68 °C for 30 s and 72 °C for 90 s. Ligated products were amplified in the same tube by adding two outermost primers (HsaGPI_F and HsaGPI_R) with 20 cycles of PCR reaction: 98 °C for 30 s, 70 °C for 30 s and 72 °C for 90 s. The *E. coli* Δ*argH* strain was constructed by substituting the *argH* of *E. coli* strain K-12 substrain MG1655 with the *npt-II* kanamycin-resistance cassette (amplified from pKD13) using lambda recombination. The kanamycin cassette was subsequently removed by flippase-mediated recombination with a helper plasmid pCP20 (ref. 54). Q5 High-Fidelity DNA Polymerase (NEB) was used to perform PCR as instructed by the manufacturer. Primer sequences are summarized in Supplementary Table 4. Successful recombinants were screened by *sacB* counterselection on salt-free Luria–Bertani (LB)-agar plate (1% tryptone, 0.5% yeast extract and 1.5% agar) containing 10% sucrose. Insertion and mutations of the human GPI gene were confirmed by Sanger sequencing. Cells were propagated in LB broth (1% tryptone, 0.5% yeast extract and 1% NaCl). For growth assays, M9 glucose medium (4 g l$^{-1}$ glucose, 47.75 mM Na$_2$HPO$_4$, 22.04 mM KH$_2$PO$_4$, 8.56 mM NaCl, 18.70 mM NH$_4$Cl, 2 mM MgSO$_4$, 0.1 mM CaCl$_2$ and trace elements) was used. Trace elements were prepared in 2,000× concentrated solution (100 mM FeCl$_3$, 9.54 mM ZnCl$_2$, 8.41 mM CoCl$_2$, 8.27 mM Na$_2$MoO$_4$, 0.75 mM CaCl$_2$, 0.91 mM CuCl$_2$ and 0.5 mM H$_3$BO$_3$ in 3.7% (w/w) hydrochloric acid solution). Kanamycin (50 µg ml$^{-1}$) was added when screening humanized *E. coli* for GPI. To

screen humanized *E. coli* for G6PD, 50 µg ml$^{-1}$ kanamycin, 100 µg ml$^{-1}$ carbenicillin and 1 mM isopropyl β-ᴅ-1-thiogalactopyranoside (IPTG) were added to media. To screen humanized *E. coli* for ASL, 100 µg ml$^{-1}$ carbenicillin and 0.01 mM IPTG were added to media. Growth profiles of *E. coli* strains were monitored using a Tecan Infinite 200 Pro microplate reader (Tecan) operated by i-Control software (v3.7.3.0, Tecan). Cells were incubated in 96-well microplates at 37 °C with a culture volume of 100 µl. Microplates were sealed with Breathe-Easy gas-permeable sealing membranes (Diversified Biotech). Optical density ($A_{600nm}$) was monitored every 15–60 min.

### Cloning G6PD variants and construction of humanized G6PD *E. coli* model
The human *G6PD* gene was chemically synthesized (IDT, sequence available in Supplementary Note 4) and cloned into the pTrcHis2A plasmid (Invitrogen) using Gibson assembly (NEBuilder HiFi DNA Assembly Kit, NEB). In brief, 5 fmol pTrcHis2A plasmid backbone (linearized by PCR) and 20 fmol *G6PD* gene fragment were mixed in 6 µl reaction followed by incubation at 50 °C for 15 min. Different G6PD variants were constructed by assembling split human *G6PD* fragments (upstream and downstream of the variation) amplified by primers containing genetic variations. Full *G6PD* constructs were assembled by OE PCR as follows: 5 ng each of the fragments were ligated by 15 cycles of the following PCR reaction: 98 °C for 30 s, 68 °C for 30 s and 72 °C for 90 s. Ligated products were amplified in the same tube by adding two outermost primers (HsaG6PD_F and HsaG6PD_R) with 20 cycles of PCR reaction: 98 °C for 30 s, 67 °C for 30 s and 72 °C for 90 s. The assembled constructs were cloned into pTrcHis2A using the aforementioned method. Then, plasmid carrying each *G6PD* variant was introduced into the *E. coli* 20.71 Δ*pgi* Δ*zwf* double-knockout strain to construct humanized G6PD *E. coli* model. Glycolytic flux was solely supported by the HMP shunt in *E. coli* 20.71 Δ*pgi* strain or 20.71 Δ*pgi* Δ*zwf* having human *G6PD*, which induces a slow growth rate, as reported elsewhere[55]. Thus, we monitored the growth of these strains for up to 60 h. Primer sequences are summarized in Supplementary Table 4.

### Measuring effects of small molecules using LEICA
The G6PD activator AG1, RRx-001, BMN, 6AN and PPC were purchased from MedChemExpress. Tc hydrochloride, MMZ and G6PDi-1, were purchased from Sigma-Aldrich. DHEA was bought from ApexBio. RRx-001 was prepared as a 100 mM dimethyl sulfoxide (DMSO) solution. AG1, G6PDi-1 and DHEA were prepared as 30 mM DMSO solutions. BMN, PPC, MMZ and 6AN were prepared as 10 mM aqueous solutions. Tc was prepared as a 10 mg ml$^{-1}$ aqueous solution. Compounds were treated at appropriate concentrations with a final DMSO concentration of 1%. DMSO was treated (1%) as an untreated vehicle control.

### Drug library screening
Cellular Metabolism Screening Library (Cayman Chemical, cat. no. 33705, batch no. 0609421) was used for drug library screening. In total, 1.2 µl of 10 mM compound solutions (DMSO) were dispensed to 96-well microplates. *E. coli* 20.71 Δ*pgi* and *E. coli* 20.71 Δ*pgi* Δ*zwf* carrying pTrc_G6PD-WT was incubated in 3 ml LB medium containing 50 µg ml$^{-1}$ kanamycin, 25 µg ml$^{-1}$ chloramphenicol and 1 mM IPTG. Then, 100 µg ml$^{-1}$ carbenicillin was added to cells carrying plasmid. One millilitre of overnight cultures was washed with 1 ml of M9 glucose medium to remove excess nutrients of LB. The washed resuspensions were inoculated into 3 ml fresh M9 glucose medium (with an initial optical density of 0.03) containing 50 µg ml$^{-1}$ kanamycin, 25 µg ml$^{-1}$ chloramphenicol and 1 mM IPTG. Then, 100 µg ml$^{-1}$ carbenicillin was added to cells carrying plasmid. Afterwards, cultures were Incubated at 37 °C for 3 h, and 120 µl of cultures were transferred to 96-well microplates having compounds resulting in 100 µM final concentration. Each plate comprises three replicated cultures of 15 compounds and vehicle control (1% DMSO) for both *E. coli* 20.71 Δ*pgi* and *E. coli* 20.71 Δ*pgi* Δ*zwf*

carrying pTrc_G6PD-WT. Cell growth was monitored every 60 min in a BioTek LogPhase 600 microplate reader (Agilent; set at 37 °C with 800 rpm shaking) operated by LogPhase 600 App (v1.08, Agilent).

### Cloning ASL variants and construction of humanized ASL *E. coli* model

The human *ASL* ORF clone was obtained from GenScript (clone ID OHu22033, NCBI reference sequence accession number NM_001024943.2) and cloned into the pTrcHis2A plasmid using Gibson assembly. In brief, 5 fmol pTrcHis2A plasmid backbone (linearized by PCR) and 20 fmol *ASL* gene fragments were mixed in 6 µl reaction followed by incubation at 50 °C for 15 min. Different *ASL* variants were constructed by assembling split human *ASL* fragments (upstream and downstream of the variation) amplified by primers containing genetic variations. Full *ASL* constructs were assembled by OE PCR as follows: 5 ng each of the fragments were ligated by 15 cycles of the following PCR reaction: 98 °C for 30 s, 68 °C for 30 s and 72 °C for 90 s. Ligated products were amplified in the same tube by adding two outermost primers (HsaASL_F and HsaASL_R) with 20 cycles of PCR reaction: 98 °C for 30 s, 67 °C for 30 s and 72 °C for 90 s. The assembled constructs were cloned into pTrcHis2A using the aforementioned method. Primer sequences are summarized in Supplementary Table 4.

### Statistical analysis

To compare the difference of means, a two-sided Welch's *t*-test was used with Bonferroni correction for multiple hypothesis testing. The dose–response curve was fitted to the Hill curve to estimate $EC_{50}$ or $IC_{50}$. For drug library screening, 15 compounds were first compared with untreated control using a two-sided Welch's *t*-test corrected for multiple hypothesis testing using the Bonferroni method. Compounds that induced a significant growth difference ($P < 0.001$) in *E. coli* 20.71 Δ*pgi* Δ*zwf* carrying pTrc_G6PD-WT strain while not inducing a significant difference ($P \geq 0.001$) for *E. coli* 20.71 Δ*pgi* were selected as primary hits.

### Reporting summary

Further information on research design is available in the Nature Portfolio Reporting Summary linked to this article.

## Data availability

The main data supporting the results in this study are available within the Article and its Supplementary Information. Allele frequencies of single-nucleotide polymorphisms in human populations were obtained from the Genome Aggregation Database (gnomAD v4.1.0) accessed through the gnomAD browser (https://gnomad.broadinstitute.org/). All compounds in the drug library ($n = 160$) are anonymized using proxy identifiers. For transparency, the actual chemical names of all compounds have been provided to the journal's editorial team in a separate document. Researchers interested in reproducing or building on this work may contact the corresponding author to request access to the compound names and structures, which will be made available under a confidentiality agreement to support ongoing research and development efforts. Full disclosure of the identity of the candidate lead compounds is deferred to a later date. Source data are provided with this paper.

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

## Acknowledgements

This work was funded by the Y.C. Fung Endowed Chair in Bioengineering at UC San Diego to B.O.P. We thank A. D'Alessandro (University of Colorado), A. Bordbar and I. Famili (Sinopia Bioscience) for fruitful discussions. We thank M. Abrams for editing the paper.

## Author contributions

B.O.P. conceived and supervised the study. D.C. and B.O.P. designed the experiments. D.C. performed the experiments. D.C. and B.O.P. analysed the data and wrote the paper. Both authors read and approved the final paper.

## Competing interests

The authors declare no competing interests.

## Additional information

**Extended data** is available for this paper at https://doi.org/10.1038/s41551-025-01391-y.

**Correspondence and requests for materials** should be addressed to Bernhard O. Palsson.

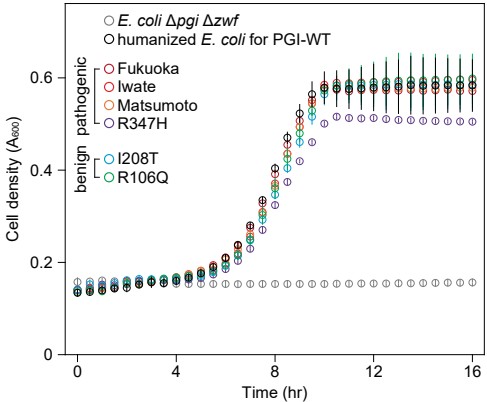

**Extended Data Fig. 1 | Growth profiles of *E. coli* carrying human glucose-6-phosphate isomerase (GPI) or its mutants in M9 glucose medium.** WT: wild-type human GPI. Data are presented as mean values +/− SD. Error bars indicate SD of ten replicated cultures.

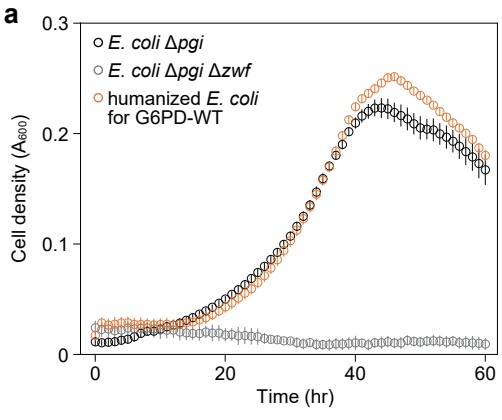

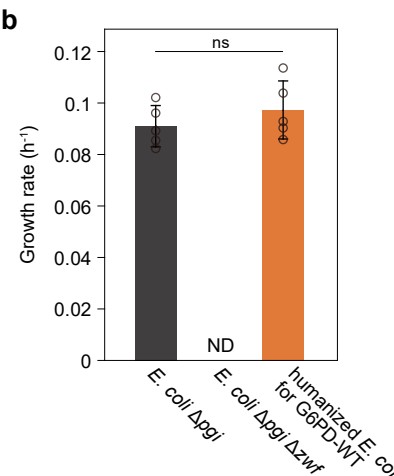

**Extended Data Fig. 2 | Growth of *E. coli* lacking phosphoglucose isomerase (*pgi*) or glucose-6-phosphate dehydrogenase (*zwf*) in M9 glucose medium.** **a**, Growth profiles of *E. coli Δpgi* and *Δpgi Δzwf* strains in M9 glucose medium. Human glucose-6-phosphate dehydrogenase (G6PD) rescued glucose utilization of *Δpgi Δzwf* knockout strain. Data are presented as mean values +/− SD. Error bars show SD of five replicates. **b**, Growth rates of *Δpgi* and *Δpgi Δzwf* strains.

*E. coli* utilizing glucose with its endogenous G6PD (Zwf; *E. coli Δpgi*) or human G6PD exhibited no significant difference in growth rate. Data are presented as mean values +/− SD. Error bars show SD of five replicates. ND, no growth was detected. ns, no significant statistical difference ($p$-value = 0.346, two-tailed Welch's $t$-test). Dots represent individual data points.

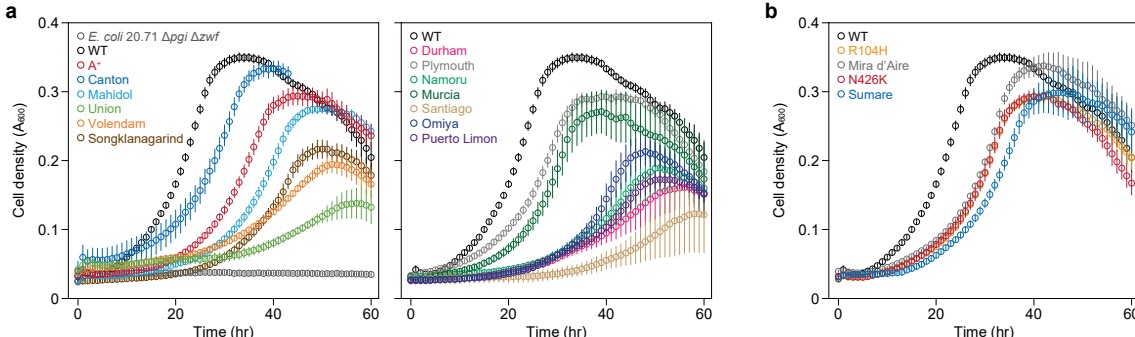

**Extended Data Fig. 3 | LEICA monitors activities of G6PD variants as growth rates are contingent on G6PD activity. a**, Growth profiles of humanized *E. coli* for G6PD expressing well-characterized variants. **b**, Growth profiles of humanized *E. coli* for G6PD expressing or less-characterized variants. Data are presented as mean values +/− SD. Error bars show SD of five replicates.

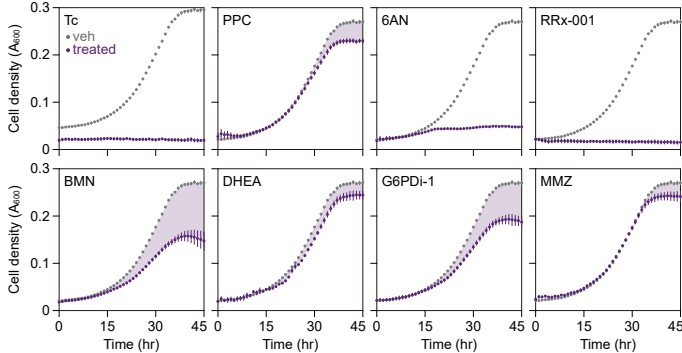

**Extended Data Fig. 4 | Effect of various compounds on humanized *E. coli* for G6PD.** Growth profiles of humanized *E. coli* for G6PD (*E. coli* 20.71 Δ*pgi* Δ*zwf* + *HsaG6PD*) strain under different drug treatments. veh: untreated control (1% DMSO). Tc: tetracycline. PPC: proparacaine. 6AN: 6-aminonicotinamide. RRx-001: bromonitrozidine. BMN: brimonidine. DHEA: dehydroepiandrosterone. G6PDi-1: inhibitor of G6PD 1. MMZ: metamizol. All compounds were treated with the final concentration of 100 μM, except for tetracycline (10 μg/ml). Data are presented as mean values +/− SD. Error bars show SD of three replicates.

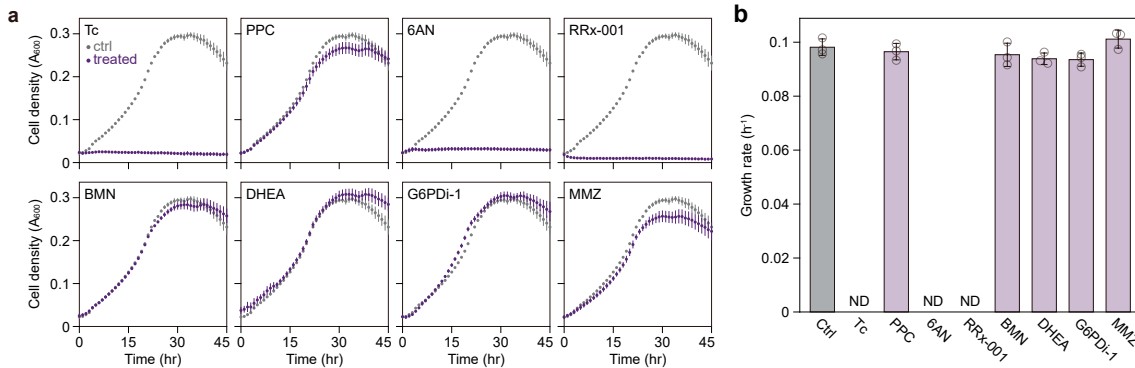

**Extended Data Fig. 5 | Effect of various compounds on *E. coli*. a**, Growth profiles of *E. coli pgi* knockout strain under different drug treatments. veh: untreated control (1% DMSO). Tc: tetracycline. PPC: proparacaine. 6AN: 6-aminonicotinamide. RRx-001: bromonitrozidine. BMN: brimonidine. DHEA: dehydroepiandrosterone. G6PDi-1: inhibitor of G6PD1. MMZ: metamizol. All compounds were treated with the final concentration of 100 μM, except

for tetracycline (10 μg/ml). Data are presented as mean values +/− SD. Error bars show SD of three replicates. **b**, Growth rates of the strain treated with the treatments. Data are presented as mean values +/− SD. Error bars show SD of three replicates. Dots represent individual data points. No significant difference in growth rate was observed (two-tailed Welch's *t*-test with Bonferroni correction).

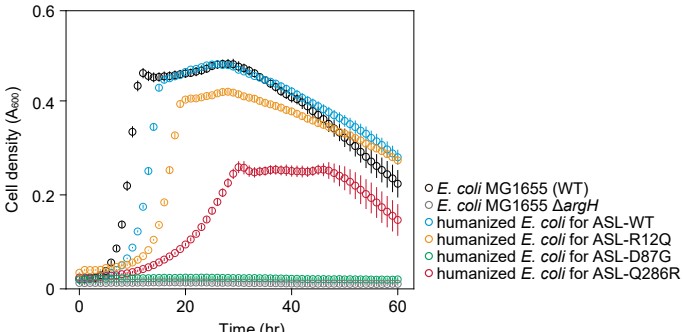

**Extended Data Fig. 6 | Growth profiles of wild-type (WT)** *E. coli*, *argH* **knockout strain, and** *argH* **knockout strain expressing WT and variants of human ASL.** Data are presented as mean values +/− SD. Error bars show SD of five replicates.

# Reporting Summary

## Statistics

For all statistical analyses, confirm that the following items are present in the figure legend, table legend, main text, or Methods section.

| n/a | Confirmed | |
|---|---|---|
| ☐ | ☒ | The exact sample size (*n*) for each experimental group/condition, given as a discrete number and unit of measurement |
| ☐ | ☒ | A statement on whether measurements were taken from distinct samples or whether the same sample was measured repeatedly |
| ☐ | ☒ | The statistical test(s) used AND whether they are one- or two-sided *Only common tests should be described solely by name; describe more complex techniques in the Methods section.* |
| ☒ | ☐ | A description of all covariates tested |
| ☐ | ☒ | A description of any assumptions or corrections, such as tests of normality and adjustment for multiple comparisons |
| ☐ | ☒ | A full description of the statistical parameters including central tendency (e.g. means) or other basic estimates (e.g. regression coefficient) AND variation (e.g. standard deviation) or associated estimates of uncertainty (e.g. confidence intervals) |
| ☐ | ☒ | For null hypothesis testing, the test statistic (e.g. *F*, *t*, *r*) with confidence intervals, effect sizes, degrees of freedom and *P* value noted *Give P values as exact values whenever suitable.* |
| ☒ | ☐ | For Bayesian analysis, information on the choice of priors and Markov chain Monte Carlo settings |
| ☒ | ☐ | For hierarchical and complex designs, identification of the appropriate level for tests and full reporting of outcomes |
| ☐ | ☒ | Estimates of effect sizes (e.g. Cohen's *d*, Pearson's *r*), indicating how they were calculated |

*Our web collection on statistics for biologists contains articles on many of the points above.*

## Software and code

Policy information about availability of computer code

| | |
|---|---|
| Data collection | Operation of microplate readers and optical density measurements were done with i-Control software (v3.7.3.0, Tecan) or LogPhase 600 App (v1.08, Agilent). |
| Data analysis | Microsoft Excel for Microsoft 365 (v2401) was used for basic analysis. Half-maximal effective concentrations were calculated by fitting data points to Hill-curves (four parameters) using Scipy package (v1.9.1, scipy.optimize.curve_fit function). |

For manuscripts utilizing custom algorithms or software that are central to the research but not yet described in published literature, software must be made available to editors and reviewers. We strongly encourage code deposition in a community repository (e.g. GitHub). See the Nature Portfolio guidelines for submitting code & software for further information.

## Data

Policy information about availability of data

All manuscripts must include a data availability statement. This statement should provide the following information, where applicable:
- Accession codes, unique identifiers, or web links for publicly available datasets
- A description of any restrictions on data availability
- For clinical datasets or third party data, please ensure that the statement adheres to our policy

The main data supporting the results in this study are available within the paper and its Supplementary Information. Allele frequencies of SNPs in human population were obtained from the Genome Aggregation Database (gnomAD v4.1.0) accessed through gnomAD browser (https://gnomad.broadinstitute.org/). All compounds

in the drug library (n=160) are anonymized using proxy identifiers. For transparency, the actual chemical names of all compounds have been provided to the journal's editorial team in a separate document. Researchers interested in reproducing or building on this work may contact the corresponding author to request access to the compound names and structures, which will be made available under a confidentiality agreement to support ongoing research and development efforts. Full disclosure of the identity of the candidate lead compounds is deferred to a later date. Source data are provided with this paper.

# Research involving human participants, their data, or biological material

Policy information about studies with <u>human participants or human data</u>. See also policy information about <u>sex, gender (identity/presentation), and sexual orientation</u> and <u>race, ethnicity and racism</u>.

| | |
|---|---|
| Reporting on sex and gender | This study did not involve human research participants. |
| Reporting on race, ethnicity, or other socially relevant groupings | N/A |
| Population characteristics | N/A |
| Recruitment | N/A |
| Ethics oversight | N/A |

Note that full information on the approval of the study protocol must also be provided in the manuscript.

# Field-specific reporting

Please select the one below that is the best fit for your research. If you are not sure, read the appropriate sections before making your selection.

☒ Life sciences　　　☐ Behavioural & social sciences　　　☐ Ecological, evolutionary & environmental sciences

For a reference copy of the document with all sections, see nature.com/documents/nr-reporting-summary-flat.pdf

# Life sciences study design

All studies must disclose on these points even when the disclosure is negative.

| | |
|---|---|
| Sample size | Experiments were performed in five or more biological replicates (except for the initial screening of drugs using LEICA; associated with Fig. 2b, 3b, which used three replicates). Sample sizes were chosen to meet or exceed the standards of previously published studies on similar subject matters (J. Biomol. Screen. 2014 19:1362-1371, Nat. Commun. 2018 9:4045, Nat. Chem. Biol. 2020 16:731-739). |
| Data exclusions | No data was excluded. |
| Replication | All experiments were replicated with three or more biological replicates. |
| Randomization | All independent biological replicates were treated identically. Thus, randomization is not applicable in this study. |
| Blinding | All independent biological replicates were treated identically. Thus, blinding is not applicable in this study. |

# Reporting for specific materials, systems and methods

We require information from authors about some types of materials, experimental systems and methods used in many studies. Here, indicate whether each material, system or method listed is relevant to your study. If you are not sure if a list item applies to your research, read the appropriate section before selecting a response.

## Materials & experimental systems

| n/a | Involved in the study |
|---|---|
| ☒ | ☐ Antibodies |
| ☒ | ☐ Eukaryotic cell lines |
| ☒ | ☐ Palaeontology and archaeology |
| ☒ | ☐ Animals and other organisms |
| ☒ | ☐ Clinical data |
| ☒ | ☐ Dual use research of concern |
| ☒ | ☐ Plants |

## Methods

| n/a | Involved in the study |
|---|---|
| ☒ | ☐ ChIP-seq |
| ☒ | ☐ Flow cytometry |
| ☒ | ☐ MRI-based neuroimaging |

## Plants

**Seed stocks**

*Report on the source of all seed stocks or other plant material used. If applicable, state the seed stock centre and catalogue number. If plant specimens were collected from the field, describe the collection location, date and sampling procedures.*

**Novel plant genotypes**

*Describe the methods by which all novel plant genotypes were produced. This includes those generated by transgenic approaches, gene editing, chemical/radiation-based mutagenesis and hybridization. For transgenic lines, describe the transformation method, the number of independent lines analyzed and the generation upon which experiments were performed. For gene-edited lines, describe the editor used, the endogenous sequence targeted for editing, the targeting guide RNA sequence (if applicable) and how the editor was applied.*

**Authentication**

*Describe any authentication procedures for each seed stock used or novel genotype generated. Describe any experiments used to assess the effect of a mutation and, where applicable, how potential secondary effects (e.g. second site T-DNA insertions, mosiacism, off-target gene editing) were examined.*

