## [Peer Review File · Nature Biomedical Engineering]

A live bacteria enzyme assay for identification of human disease mutations and drug screening

Corresponding Author: Prof Bernhard Palsson

Version 0:

Decision Letter:

Dear Dr Palsson,

Thank you again for submitting to *Nature Biomedical Engineering* your manuscript, "A live bacteria enzyme assay facilitates rapid identification of mutations associated with human genetic diseases and drug screening". The manuscript has been seen by three experts, whose reports you will find at the end of this message.

You will see that the reviewers appreciate the work. However, they articulate concerns about the translational utility of the approach, and provide useful suggestions for improvement. We hope that with substantial further effort you can address the criticisms, increase the strength of the evidence, and convince the reviewers of the merits of the study. In particular, we would expect that a revised version of the manuscript provides:

- * Robust validation of the assay.
- * Evidence of the assay's utility in the identification of less-characterized gene variants.
- * Evidence of the assay's utility in a translationally relevant setting.

When you are ready to resubmit your manuscript, please upload the revised files, a point-by-point rebuttal to the comments from all reviewers, the [reporting summary](https://www.nature.com/authors/policies/ReportingSummary.pdf), and a cover letter that explains the main improvements included in the revision and responds to any points highlighted in this decision.

Please follow the following recommendations:

- * Clearly highlight any amendments to the text and figures to help the reviewers and editors find and understand the changes (yet keep in mind that excessive marking can hinder readability).
- * If you and your co-authors disagree with a criticism, provide the arguments to the reviewer (optionally, indicate the relevant points in the cover letter).
- * If a criticism or suggestion is not addressed, please indicate so in the rebuttal to the reviewer comments and explain the reason(s).
- * Consider including responses to any criticisms raised by more than one reviewer at the beginning of the rebuttal, in a section addressed to all reviewers.
- * The rebuttal should include the reviewer comments in point-by-point format (please note that we provide all reviewers will the reports as they appear at the end of this message).
- * Provide the rebuttal to the reviewer comments and the cover letter as separate files.

We expect that you will be able to resubmit the manuscript within 16 weeks of receiving this message. If this is the case, you will be protected against potential scooping. Otherwise, we will be happy to consider a revised manuscript as long as the significance of the work is not compromised by work published elsewhere or accepted for publication at *Nature Biomedical Engineering*.

We hope that you will find the referee reports helpful when revising the work. Please do not hesitate to contact me should

you have any questions.

Best wishes,

Pep

Pep Pàmies

Chief Editor, Nature Biomedical Engineering

Reviewer #1 (Report for the authors (Required)):

Summary: This original research article presents a simple and rapid method for investigating human enzymopathy through the growth rate measurement of *E. coli*, expressing human mutant enzymes. The replacement of *E. coli* genes with their human orthologs is performed according to a previously published method by the same research group (Sandberg, et al., *Nat Ecol*, 2020). By knocking out the bacterial phosphoglucose isomerase (*pgi*) and glucose-6-phosphate dehydrogenase (*zwf*) genes, glycolytic flux is forced through the recombinant human enzymes glucose-6-phosphate isomerase (GPI) or glucose-6-phosphate dehydrogenase (G6PD). Growth rate is then shown to correlate well with previously reported values of enzymatic activities for different mutations. The method is sufficiently sensitive to detect dose-dependent and mutation-dependent pharmacodynamics of selected known small-molecule activator and drug inhibitors. While the presented method is simple and elegant the manuscript in its current scope should not be considered for publication in *Nature Biomedical Engineering* due to the following major concerns:

- 1) The translational potential of the method is not sufficiently supported, as expected by the journal's standards. This may include, but is not limited to, screening genetic variants within a real clinical patient pool, and verifying inhibitor/activator pharmacodynamics via hemolysis assays, high-throughput cellular assays, or animal *in vivo* models.
- 2) The authors lightly dismiss some major analytical challenges they indicate, including differences in membrane transport between bacterial and mammalian cells, the toxicity of some studied compounds to *E. coli*, and discrepancies between the presented method and cell-based assays.
- 3) The presented work lacks the use of tools or principles from physical sciences, engineering or mathematics as required by the journal's scope. For example, how could the proposed method be incorporated within a medical device? Could it be coupled with an algorithm to diagnose mutations or prescribe a drug?

Minor concerns

-
- 1) Growth rates reported in this work are very low, specifically for the humanized *E. coli* for G6PD. This issue should be addressed and explained.

Stylistic recommendations:

-
- 1) The work could be presented more concisely
 - a. Line 91 – describes a previous study – could be shortened to one sentence
 - b. Line 103 – this paragraph does not contribute to the understanding of the results
 - c. Line 122-130 – should be included in the introduction
 - d. Line 170 – this paragraph does not contribute to the understanding of the results
 - 2) Results could be presented more concisely with better format consistency:
 - a. Figure 2a,c could be combined in Figure 1 and presented in the same format. Growth curves in Figure 2b could be in Extended data
 - b. Growth curves in Figure 4 could be in Extended data, and Figure 3 could be combined into Figure 4 and presented in the same format as all other dose-response plots (Extended data Figure 3)

Reviewer #2 (Report for the authors (Required)):

Identification of genetic variations associated with human diseases is crucial for advancing our understanding and treatment of these conditions. Genome-wide association studies (GWAS) are an effective method for uncovering these variations. However, inferring the functional impact of these variations remains a challenge, often necessitating specialized assays and time-consuming analytical methods and instrumentation.

The authors propose an innovative strategy, termed LEICA, which leverages humanized *E. coli* to study human genetic enzymopathies and to screen candidate drugs targeting metabolic pathways. Specifically, they focus on GPI and G6PD, which are implicated in common hereditary enzymopathies. By substituting selected *E. coli* metabolic enzymes with human

orthologs and their variants, they demonstrate that the growth rate of *E. coli* reflects the in vivo activity of heterologously expressed human enzymes. This method enables high-throughput screening of drug effects on human enzymes and bridges the gap between in vitro assays and cell-based assays. The use of live bacterial cells for enzyme activity assessment offers a rapid and accurate means of identifying the impact of genetic mutations, providing valuable insights into the causality of genetic disorders.

The authors also suggest that LEICA holds potential for personalized medicine, offering a cost-effective platform for identifying drug sensitivities and developing targeted therapies. The growth rate of *E. coli*, which is dependent on glycolytic flux or rate-limiting steps, serves as a proxy for the activity of human enzymes expressed in the bacteria. *E. coli* strains carrying different enzyme mutants displayed distinct growth rates, reflecting variations in enzyme activity induced by mutations. This approach eliminates the need for laborious protein purification steps typical of traditional in vitro assays.

Several concerns need to be addressed:

Assessment of Catalytic Activity Relevance: On line 206, the authors state, "Human genetic disorders result from a complex interplay of biological factors, including variations in catalytic activity, expression levels, and zygosity, posing challenges in identifying causality. LEICA insulates enzyme activity from such biological factors by mimicking the homozygous state and providing stable expression levels, offering a rapid means of assessing only catalytic activity as a variable." While this statement is well-founded, I question the overall value of developing a technology like LEICA that correlates catalytic activity with cell growth. Given the complex interplay of factors in human genetic disorders, LEICA's application may be limited to well-characterized variants. If so, what is the utility of LEICA in studying and diagnosing less-characterized variants? Could it risk complicating interpretations rather than simplifying traditional assays? I suggest that the authors include genetic variants with limited background information for this study.

Validation of Variants: Although the authors demonstrate that GPI and G6PD variant activities correlate strongly with *E. coli* growth rates, the sample size ($n = 6$) is too small to robustly validate the hypothesis or generalize the findings. A larger number of GPI and G6PD variants should be tested to strengthen the correlation and provide more comprehensive data.

Screening of Small Molecules: While LEICA appears promising for screening small molecules like AG1, there is skepticism about its effectiveness for screening inhibitors. Given that LEICA might also select compounds with antibacterial properties or effects on primary metabolism, it may result in many false positives when screening for G6PD inhibitors. To validate LEICA's utility for drug screening, it is recommended that the authors conduct screens using e.g., commercially available compound libraries to evaluate its efficacy in identifying both activators and inhibitors.

Application to Personalized Medicine: This manuscript focuses on validating LEICA with previously known data. However, it does not demonstrate how LEICA could provide new insights or expand our understanding. I question whether LEICA genuinely contributes valuable information for personalized medicine.

Overall, while LEICA is an intriguing approach with potential applications, the study would benefit from addressing these concerns to enhance its robustness and applicability.

Reviewer #3 (Report for the authors (Required)):

Choe and Palsson report an approach for high-throughput screening of candidate drugs in a humanized *E. coli* platform (LEICA) in an effort to develop a precision medicine approach to human genetic disorders. The relationship between a detected genetic variant and its potential functional significance is a major issue in the practice of medicine. By replacing *E. coli* glycolytic enzymes with their human equivalents, the growth rate of *E. coli* was found to correlate highly with previously reported activities of recombinant enzymes. By incorporating genes coding for human glucose-6-phosphate isomerase (G6PD) and GPI into the LEICA system, human enzyme activities can be measured in an intracellular environment that provides the physiological milieu for proper enzyme function. This system offers a relatively straightforward and rapid way to measure human enzyme activity compared to current specialized assays of enzyme activities. Such a screening technique could be adapted to test small molecules for their potential to act as therapies for specific genetic enzymopathies.

Focusing on six representative mutations (2 benign, 4 pathogenic) in the gene coding for GPI, the authors showed that strains carrying different GPI variants showed different growth rates compared to the wild-type humanized *E. coli* for GPI. Strains harboring pathogenic variants showed significantly lower growth rates; benign variants caused weakly impaired or no change in growth rate compared to the wild type control. In short, LEICA can be used to estimate activities of a human enzyme.

Studies focusing on human glucose 6-phosphate dehydrogenase (G6PD), showed that this enzyme is also functional in the *E. coli* system. Similarly to the GPI studies, differential growth, related to the pathogenic severity of the underlying genetic alteration, was seen in the LEICA system when different G6PD gene variants were studied.

The LEICA system also exhibited potential as a drug-screening platform, as the use of a small molecule activator of G6PD activity (AG1) resulted in growth rates of humanized strains for various G6PD enzymes that correlated to known activation profiles reported in other assay systems. In addition, seven other compounds with documented effects on G6PD activity were studied. LEICA generally generated results similar to cell-based recombinant assays versus in vitro assays and was considered to be a suitable platform for high-throughput screening. This system also has potential as a precision medicine

tool, as the studied G6PD genetic variants showed different responses to each tested compound.

The manuscript is clearly written and background related to the studied enzymes is provided to assist with understanding these studies in context.

The LEICA system appears to be robust in interrogating enzymes related to glycolysis. Given the interest in finding novel therapies for inborn errors of metabolism, such as PKU, MSUD, urea cycle disorders, organic acidemias, fatty acid oxidation disorders, etc. this platform would appear to have multiple potential applications. Do the authors have thoughts related to how this *E. coli* system might perform while evaluating other enzymes related to human disease in different biochemical pathways, e.g., those involving small molecules, such as amino acid or fatty acid metabolism, or organelle disorders, such as those involving lysosomes, peroxisomes, or mitochondria?

Minor comment:

Gene names should be italicized.

Version 1:

Decision Letter:

Dear Bernhard,

Thank you for your revised manuscript, "A live bacteria enzyme assay facilitates rapid identification of mutations associated with human genetic diseases and drug screening". Having consulted with the original reviewers (whose comments you will find at the end of this message), I am pleased to write that we shall be happy to publish the manuscript in *Nature Biomedical Engineering*.

We will be performing detailed checks on your manuscript, and in due course will send you a checklist detailing our editorial and formatting requirements. You will need to follow these instructions before you upload the final manuscript files.

Best wishes,

Pep

Pep Pàmies

Chief Editor, *Nature Biomedical Engineering*

Reviewer #1 (Report for the authors (Required)):

Summary: The impact of the work has been significantly enhanced through the extensive inclusion of the following findings:

- Identification of seven potential lead compounds for G6PD inhibition within a 160-member drug library, with four of the hits being novel discoveries.
- Analysis of four less-characterized G6PD variants with no previously reported biochemical properties.
- Demonstration of a linear correlation between the growth rates of humanized *E. coli* expressing G6PD and catalytic constants (k_{cat}) for 13 additional well-characterized variants.
- Expansion of LEICA beyond glycolysis to screen the activity of human argininosuccinate lyase (ASL).

The revised manuscript now more effectively highlights the translational potential of this straightforward and cost-effective method for screening genetic variants within a real clinical patient pool.

The analytical limitations of the method are now more thoroughly addressed, with a detailed comparison to other gold-standard methods. The authors have clarified the specific application of LEICA for investigating enzyme activity independently of other pathogenic factors, emphasizing both its advantages and drawbacks.

All inquiries were carefully addressed and clarified, and all stylistic suggestions were implemented.

Optional comment: Regarding the drug library screening, did you encounter any false-negative results? Specifically, were there compounds known to inhibit G6PD but not identified as such by LEICA?

Reviewer #2 (Report for the authors (Required)):

All of my concerns have been thoroughly addressed, and I now recommend acceptance.

Reviewer #3 (Report for the authors (Required)):

Since the initial submission, the authors have performed additional experiments and have made significant additions to the manuscript. In addition to expanding the work related to glycolytic enzymes, argininosuccinate lyase (ASL), an essential urea cycle enzyme, was also studied in the LEICA system. Interestingly, humanized *E. coli* expressing pathogenic ASL variants showed markedly reduced growth rates compared to wild-type cells. I appreciate this addition, as the LEICA method has now shown applicability to metabolic pathways related to amino acid metabolism. As there are numerous biochemical genetic disorders related to impaired to the metabolism of amino acids (and of course other substrates), I look forward to following the results of this screening assay as it relates to an expanding number of conditions.

Version 2:

Decision Letter:

Dear Prof Palsson,

I am happy to inform you that your manuscript, "A live bacteria enzyme assay for identification of human disease mutations and drug screening", has now been accepted for publication in *Nature Biomedical Engineering*.

Over the next few weeks, the figures will be checked for production quality, the text edited to ensure that it conforms to house style, and the manuscript typeset.

Our Articles are published about 40 days after the acceptance date (we recommend that you inform your institutional press office of this timeframe), and you will be notified of the actual publication date a few days in advance. Articles can be published any working day of the week, and are pushed live shortly after 10 am London time.

Publishing agreement. You will be asked to digitally sign a publishing agreement (grant of rights). After the signed publishing agreement has been received, the proofs of the article will be sent to you for review. If you have any queries during the production process, or you cannot meet the requested deadline for returning the proofs, please contact rjsproduction@springernature.com.

Nature Biomedical Engineering is a Transformative Journal. Authors may publish their research with us through the traditional subscription access route, or make their paper immediately open access through payment of an article-processing charge. More [information about publication options](https://www.springernature.com/gp/open-research/transformative-journals) is available.

You may need to take specific actions to [comply](https://www.springernature.com/gp/open-research/funding/policy-compliance-faqs) with funder and institutional open-access mandates. If the work described in the accepted manuscript is supported by a funder that requires immediate open access (as outlined, for example, by [Plan S](https://www.springernature.com/gp/open-research/plan-s-compliance)) and your manuscript was originally submitted on or after January 1st 2021, then you should select the gold OA route. Authors selecting subscription publication will need to accept our standard licensing terms (including our [self-archiving policies](https://www.springernature.com/gp/open-research/policies/journal-policies)), and these will supersede any other terms that the author or any third party may assert apply to any version of the manuscript.

Acceptance of your manuscript is conditional on agreement, by all authors, with both our [media embargo](http://www.nature.com/authors/policies/embargo.html) and [confidentiality and pre-publicity](http://www.nature.com/authors/policies/confidentiality.html) policies. In particular, you may arrange your own publicity of the Article (for instance, through your institutional press office), as long as you ensure that journalists strictly adhere to the media embargo.

To assist you in disseminating the work, as soon as the Article is published you will be able to take advantage of the Springer Nature [SharedIt](https://www.springernature.com/gp/researchers/sharedit) initiative to [generate a unique shareable link to the Article](http://authors.springernature.com/share) that will allow anyone (with or without a subscription) to read it. Recipients of the link who are subscribers will also be able to download and print the PDF.

Thank you for having submitted this work to *Nature Biomedical Engineering*.

Best wishes,

Barbara Cheifet

**A live bacteria enzyme assay facilitates rapid identification of mutations**
**associated with human genetic diseases and drug screening**

Donghui Choe¹ and Bernhard O. Palsson^{1,2*}

¹Shu Chien-Gene Lay Department of Bioengineering, University of California San Diego, La
Jolla, CA 92093, USA

²Department of Pediatrics, University of California, San Diego, La Jolla, CA 92093, USA

*Correspondence should be addressed to B.O.P. (bpalsson@ucsd.edu)

**Keywords:** Live *Escherichia coli* assay (LEICA), humanized *E. coli*, *in vivo* screening system

**ABSTRACT**

Advances in high-throughput sequencing have enabled the identification of genetic variations
associated with human disease. However, deciphering the functional significance of these
variations remains **challenging**. Here, we propose an alternative approach **that utilizes**
humanized *Escherichia coli* to study human genetic enzymopathies and to screen candidate
drug effects on metabolic targets. By replacing selected *E. coli* metabolic enzymes with their
human orthologs and their sequence variants, we demonstrate that the growth rate of *E. coli*
reflects the *in vivo* activity of heterologously expressed human enzymes. This approach
accurately reflected enzyme activities of known sequence variants, thus enabling **rapid**
screening of **causal sequence variations associated with human diseases**. This approach
bridges the gap between *in vitro* assays and cell-based assays. Our findings suggest that the
proposed approach using a humanized *E. coli* strain holds promise for **drug discovery**, offering
a **high-throughput** and cost-effective platform for identifying **new compounds targeting human**
**enzymes**. Continued research and innovation in this field have the potential to impact the
development and practice of precision medicine.

INTRODUCTION

The human genome sequence varies between individuals. Everyone typically harbors
between 250 to 300 loss-of-function sequence variations relative to the reference human
genome¹. These sequence variations often underlie inherited disorders, making the
identification of causal mutations an **essential** pursuit in human biology and personalized
medicine. This imperative is underscored by recent strides in *in vivo* genome editing, which
offer promising avenues for treating conditions like eye and liver disease^{2,3}.

Traditionally, the identification of genotypes associated with genetic disease has relied on
direct patient analysis and subsequent documentation as case reports. Once genetic
variations linked to specific disorders are mapped, they are studied through various
methodologies. One common approach involves *in vitro* studies utilizing materials obtained
from patients or recombinant proteins to characterize mutant proteins. However, this approach
is constrained by sample availability and necessitates labor-intensive purification processes.
Moreover, it often requires specialized assay systems tailored to individual proteins, alongside
time-consuming analytical methods and instrumentation. **Additionally**, dilute *in vitro* conditions
do not replicate crowded and dense gel-like physiological conditions, necessitating an *in vivo*
system for an accurate biochemical assay⁴.

Alternatively, large-scale computational surveys, such as genome-wide association studies
based on population sequencing datasets⁵, have proven effective in identifying potential
associations between thousands of genetic changes to pathological conditions⁶. However,
such associations do not establish causality, particularly given the vast number of
uncharacterized genetic variations and confounding factors of zygosity and genetic linkage^{5,7}.
**Furthermore, variations that induce severe pathogenicity from early onset (such as in**
**newborns and infants) are rare in the human population, making it challenging to obtain**
**samples or detect such variations in population databases. As a result, pathogenic mutations**
**remain practically undetectable, while detected mutations are more likely to have only a minor**
**impact on enzyme activity.**

In prior work, we demonstrated the replacement of *Escherichia coli* glycolytic genes with **their**
human orthologs⁸. Despite the evolutionary distance between bacteria and humans, they
share significant metabolic similarities, such as a universal glycolytic framework across phyla,
where glucose is metabolized through identical chemical reactions, albeit with different
enzymes. **Thus**, *E. coli* lacking key native enzymes **for glucose metabolism could utilize**
**glucose with the** introduction of corresponding human enzymes without alterations to their
coding sequence.

Thus, we proposed an alternative approach to investigate the activities of human mutant
enzymes by developing a live *Escherichia coli* assay (LEICA). Specifically, we focused on
glucose-6-phosphate isomerase (GPI) and glucose-6-phosphate dehydrogenase (G6PD),
which are associated with the most common human hereditary enzymopathies^{9,10}.

As the growth rate of *E. coli* is contingent upon glycolytic flux¹¹, it serves as a surrogate
measure for the activity of heterologously expressed human enzymes. Harnessing the ease
of genetic manipulation in *E. coli*, we replicated human mutations in these enzymes. *E. coli*
strains carrying different mutants exhibited distinct growth rates, reflecting variations in
enzyme activity induced by mutations. Notably, the growth rates demonstrated a high linear
correlation with enzyme activities previously determined through *in vitro* assays with
recombinant proteins. This live bacterial cell assay provides an accurate and rapid means of
screening for enzyme activity change resulting from human mutations, offering insights into
the causality of genetic disorders attributable to genetic variations. We also expanded LEICA
to screen argininosuccinate lyase (ASL), a key urea cycle enzyme whose deficiency results in
argininosuccinic aciduria¹². Complementation of arginine auxotrophy in *E. coli* lacking ASL
with human equivalents demonstrates LEICA's broader applicability across diverse enzymes.

LEICA revealed other potential uses. Through experimentation with small molecules using
LEICA, we found that chemical compounds targeting human G6PD could either enhance or
inhibit growth when administered directly to cells in culture. We confirmed inhibitory effects of
known G6PD inhibitors, highlighting the drug screening capability of LEICA on human drug
targets. Screening a library of 160 human metabolism modulators revealed seven lead
compounds, including the rediscovery of three with known G6PD inhibitory effects or
antimalarial activity. Lastly, we observed enhanced growth of *E. coli* carrying the G6PD Canton
mutant when treated with the recently discovered G6PD agonist AG1¹³. LEICA thus serves
not only as a drug screening platform but also holds promise for use in developing
personalized medicine.

RESULTS

**Growth of *Escherichia coli* carrying human glucose-6-phosphate isomerases (GPI)**
**reflects *in vivo* enzyme activity:** In a previous study, we demonstrated the successful
replacement of *E. coli* GPI with its human ortholog⁸. During adaptive laboratory evolution to
optimize growth, we observed that achieving optimal GPI function in live bacteria did not
necessitate mutations in the protein sequence. Instead, enhanced gene expression, facilitated
by two point mutations in the promoter, proved to be the key factor for growth improvement.

Furthermore, various *in vitro* studies use recombinant human enzymes expressed from *E. coli*
lysates^{14,15}, indicating *E. coli* as a potential host for functional expression of human enzymes.

In both humans and *E. coli*, glucose is mainly utilized by glycolysis. Thus, we hypothesized
that replacing a glycolytic enzyme in *E. coli* with its human counterpart would reflect human
enzyme activity (Fig. 1a). However, glucose can also be utilized via the oxidative pentose
phosphate pathway (oxPPP), also known as hexose monophosphate (HMP) shunt. Therefore,
to specifically assess human GPI activity in *E. coli*, it is essential to disengage the HMP shunt.
This was achieved by deleting the gene for G6PD (*zwf*) that enables an alternate glucose
utilization. We thus knocked out *zwf* from the *E. coli* strain 20.71—an evolved *E. coli* K-12
MG1655 strain¹⁶ carrying in-frame *pgi* swap to human GPI and promoter mutations (Fig. 1b).
In the resulting strain—a humanized *E. coli* for GPI—growth in a glucose-containing medium
is solely dependent on GPI activity (Extended Data Fig. 1).

We sought to examine the effects of genetic variations in GPI found in the human population
that are associated with hemolytic anemia. To explore the effects of both benign and
pathogenic mutations, we selected six representative mutations (Supplementary Table 1),
some of which are supported by direct biochemical evidence¹⁴. Two of the six mutations are
known to be benign, with population frequencies ranging from 1 in 49 to 1 in 4,400 individuals
(Genome Aggregation Database; gnomAD¹⁷), while four pathogenic variations were found at
frequencies of 1 in 57,000 to 1 in 390,000 individuals (gnomAD), with some reported from
patients exhibiting hemolysis^{18–20}, indicating their disease association.

To evaluate the functionality of a live bacterial assay, we replaced wild-type (WT) human GPI
in the humanized *E. coli* for GPI with the six mutant GPIs to investigate activity differences
induced by the mutations. Strains expressing different GPI variants showed growth rate
changes ranging from -13.1 to +2.2% compared to the WT control (Fig. 1c). Growth rates of
strains carrying pathogenic mutations were significantly lower than those with WT GPI, while
two benign mutants displayed either a mild decrease in growth (-5.6%) or growth
indistinguishable from the WT control.

Comparison of biochemically determined properties of recombinant mutants¹⁴, Fukuoka,
Iwate, and Matsumoto, with the live bacterial assay revealed a high linear correlation between
enzyme activity and the growth rate. This result indicates a direct impact of GPI activity on the
growth rate of humanized *E. coli* strains (Fig. 1d). This demonstrates that a live *Escherichia*
*coli* assay (LEICA) is capable of inferring activities of human enzymes. LEICA gives an
impetus for rapidly screening genetic variants causing enzyme deficiency. Compared to

conventional biochemical assays using hemolysate from isolated red blood cells (RBCs) or
recombinant enzymes that require specialized assay systems and laborious purification
steps²¹, LEICA calls for only a single gene replacement (and variants therein) with a
straightforward growth measurement, as all the necessary substrates and cofactors are
present in live bacteria.

**Diagnosing human glucose-6-phosphate dehydrogenase (G6PD) deficiency using**
**LEICA:** Having demonstrated the concept of LEICA analyzing human enzyme activity, we
sought to explore another human enzyme, G6PD. G6PD deficiency is the most common cause
of RBC enzymopathy, affecting an estimated 300-400 million people worldwide²². G6PD plays
a critical metabolic role in RBCs as the HMP shunt is the main source of NADPH required to
maintain redox homeostasis²³. Hereditary G6PD deficiency is caused by sequence variations
in *G6PD*, which is highly polymorphic with over 230 sequence variants identified^{6,10}, due to its
association with malaria resistance²⁴. Despite the growing number of variants identified
through populational sequencing, clinical or molecular characterizations remain limited⁶.

[revised manuscript text omitted]

**Small-molecules activate or inhibit glucose 6-phosphate dehydrogenase in humanized**
***E. coli***: Human G6PD is functionally active only in a dimer or tetramer, and some mutations
have been shown to hamper oligomerization³⁶. A recently discovered small-molecule activator
of G6PD, named AG1, promotes and stabilizes dimer formation^{13,15}. Thus, it activates G6PD
mutants that have a reduced oligomerization state, e.g., G6PD-Canton^{13,15}. According to the
previous report, AG1 improved activity of the G6PD-Canton variant up to 1.7-fold with half
maximal effective concentration (EC₅₀) of 3.4 μM¹⁵. It also increased basal activity of WT G6PD

by approximately 20%¹⁵. Thus, we examined the effect of AG1 using LEICA to determine if it
could be used to screen for the effect of small molecules.

The effect of AG1 exposure on growth of *E. coli* carrying its endogenous *G6PD* (*E. coli* 20.71
Δ *pgi*) was examined. AG1 treatment showed no effect on growth, thus ruling out its effect on
endogenous *E. coli* metabolism (Fig. 2). When humanized strains for G6PDs were examined,
AG1 improved the growth rates of humanized *E. coli* expressing WT G6PD and Canton variant
up to 12.8% and 20.1% at 0.3 μ M, respectively (Fig. 2a); a result that is consistent with the
previous study (Supplementary Note 1)¹⁵. Activated G6PD-Canton supported growth of
humanized *E. coli* better than untreated WT and there was no significant difference in the
activities when maximally activated (*p*-value of 0.055; Welch's *t*-test, Fig. 2a). It is worth noting
that the effective concentrations of AG1 needed in LEICA experiments were much lower than
those of *in vitro* and cell-based screens¹⁵. The discrepancy of effective concentrations might
have originated from different concentrations of reaction constituents (i.e., substrates,
cofactors, and enzymes) in the *in vitro* reconstituted system and bioavailability of small
molecules across bacterial and mammalian cell membranes. Nonetheless, LEICA for G6PD
successfully reflected the known activation of human G6PD by small molecule activator AG1
without any specialized reporter-based assay system.

Given that LEICA can measure the effect of AG1 on human G6PD, we aimed to evaluate its
ability to screen various compounds that target G6PD. As G6PD deficient patients face a high
risk of hemolysis when exposed to certain drugs, specific foods, or infections³⁷, it is important
to evaluate the impact of drugs on G6PD when devising treatment strategies for G6PD-
deficient individuals. Additionally, drugs that act on G6PD are active areas of clinical research,
as it is linked to antimalarials, immune response, and tumors³⁸⁻⁴¹.

Seven compounds with known effects on G6PD were chosen for screening (Supplementary
Note 2). These include: (1) the local anesthetic proparacaine (PPC), (2) the non-selective
inhibitor of NADP-dependent enzymes 6-aminonicotinamide (6AN), (3) the energetic
compound, radiosensitizer RRx-001, (4) the α_2 adrenergic agonist brimonidine (BMN), (5) the
steroid hormone dehydroepiandrosterone (DHEA), (6) the small molecule G6PD inhibitor
G6PDi-1, and (7) the pyrazolonic analgesic metamizol (MMZ). Previous reports indicate that
PPC has no significant effect on G6PD⁴², while G6PDi-1⁴¹, DHEA⁴³, BMN⁴², and MMZ⁴⁴ exhibit
inhibitory effects. RRx-001 and 6AN are non-selective inhibitors^{45,46}, either generating reactive
radicals or non-selectively inhibiting NADP-dependent enzymes. Each compound was
supplemented to culture media at a concentration of 100 μ M and 10 μ g/ml tetracycline (Tc)
was used as a positive control for bacterial growth inhibition.

LEICA treated with PPC exhibited no difference in growth profile compared to the untreated
control (Fig. 2b and Extended Data Fig. 4), aligning with previous observations⁴². Treatment
with Tc, RRx-001, and 6AN completely halted the growth of humanized *E. coli* for G6PD in
LEICA (Fig. 2b). However, the growth of *E. coli* Δ *pgi* was also suppressed by RRx-001 and
6AN (Extended Data Fig. 5), indicating their toxicity to *E. coli* and prompting us to exclude
them as hits.

The growth of humanized *E. coli* for G6PD (WT) was significantly inhibited by G6PDi-1, DHEA,
and BMN (Fig. 2b,c and Extended Data Fig. 4), with no effect observed on *E. coli* Δ *pgi*
(Extended Data Fig. 5), underscoring the inhibitory effect of these compounds on human
G6PD, consistent with the previous reports^{41–43}. However, MMZ showed no discernible effect.
This is likely attributable to the high inhibitory concentration of MMZ (IC₅₀ of 17 mM) required
to inhibit G6PD activity⁴⁴, in contrast to G6PDi-1 and BMN, whose IC₅₀s are in the sub-
millimolar range (0.07-30 μ M).

Two compounds—G6PDi-1 and BMN—which exhibited inhibitory effects in the primary
screen, underwent further examination in assays with multiple doses to validate the screening
results. We also included DHEA that has been reported to have varying G6PD inhibitory effect
depending on the assay format. LEICA was performed with each compound at concentrations
ranging from 5 to 300 μ M, revealing a dose-dependent response with IC₅₀ of 111, 138, and
1309 μ M, respectively (Fig. 2c-e). These IC₅₀ values differ from those obtained in *in vitro*
assays where the IC₅₀ of G6PDi-1, DHEA, and BMN were reported as 0.07, 9, and 30 μ M,
respectively^{41,42}. However, IC₅₀ values measured from LEICA are more closely aligned with
results from cell-based assays, where the IC₅₀ value for G6PDi-1 ranged from 13 to 31 μ M
depending on the cell type. Interestingly, DHEA exhibited an inhibitory effect in LEICA,
contrary to the lack of inhibitory effect observed in cell-based assays⁴¹. Previous reports
indicate the antiproliferative effects of DHEA on various cells may stem from indirect effects
that are not fully understood⁴⁷. Thus, the behavior of LEICA in this context resembles that of
a recombinant assay. Despite these discrepancies, LEICA effectively screened compounds
with comparable IC₅₀ values, making it a suitable format for high-throughput screening
applications.

**Drug library screening identifies potential lead compounds for G6PD: Having**
**demonstrated the sensitivity of LEICA with known compounds that modulate G6PD activity,**
**we screened a drug compound library to identify new lead compounds with potential effects**
**on G6PD. The screening involved two stages: (1) a single-dose growth assay using LEICA,**

and (2) validation of primary hits through dose-response assays (Fig. 3a). Of the 160 human
metabolism modulators screened, 7 compounds (primary hits) were identified that inhibited
the growth of *E. coli* expressing human *G6PD*, while showing no effect on the strain with
endogenous *E. coli G6PD* (Fig. 3b).

The screening successfully re-discovered G6PDi-1 as a primary hit, confirming the assay's
robustness. Two other primary hits, methotrexate and gossypol, have also been previously
linked to G6PD. Methotrexate, a dihydrofolate reductase inhibitor, has reported G6PD
inhibitory effects (IC₅₀ of 114 μM)⁴⁸ and antimalarial activity⁴⁹. Gossypol, known for its
antimalarial properties, inhibits oxidoreductases, including G6PD in *Trypanosoma cruzi*⁵⁰.
Given that G6PDi-1 was originally discovered as an inhibitor of *T. cruzi* G6PD⁵¹, it logically
follows that gossypol, with its reported effects on *T. cruzi*, might also inhibit human G6PD.
These three previously known compounds detected in the screen validate the sensitivity and
reliability of the assay.

The four remaining compounds have no previously reported association with G6PD activity;
however, three completely inhibited the growth of *E. coli* expressing human G6PD. Further
characterization using dose-response assays revealed reliable dose-dependent effects from
all the primary hits (Fig 3c). Among the four newly identified compounds, three displayed
higher potency with IC₅₀s ranging from 15.6 to 109 μM, compared to G6PDi-1, methotrexate,
and gossypol (IC₅₀s of 111-148 μM). This result suggests that these newly identified
compounds are candidate leads for novel antimalarial drugs as potent G6PD inhibitors. The
drug screening demonstrates LEICA's potential as a versatile high-throughput screening tool
capable of identifying compounds that influence human enzymes within a bacterial context.

**Expanding LEICA's application for enzymes other than glycolysis:** To broaden LEICA's
applicability to enzymes outside glycolysis, we focused on argininosuccinate lyase (ASL), a
critical enzyme in the urea cycle (Fig. 4a)⁵². Although *E. coli* does not have a urea cycle, it
contains an ASL enzyme encoded by the *argH* gene, which is involved in arginine
biosynthesis. Since knockout of *argH* results in arginine auxotrophy, it creates a suitable
platform for assaying human ASL activity by growing cells in arginine-free conditions (Fig. 4b).
We found that human ASL could complement the arginine auxotrophy in an *E. coli argH*
knockout strain, indicating successful functional expression of human ASL in *E. coli*
(Extended Data Fig. 6).

We then examined various ASL sequence variants that are known to cause argininosuccinic
aciduria (ASA), with residual enzyme activity ranging from 0 to 6.09% (Supplementary Table

3)^{12,52}. The humanized *E. coli* for ASL whose arginine biosynthesis is supported by different
pathogenic ASL variants exhibited significant growth reductions (ranging from 16 to 100%)
compared to cells expressing WT human ASL (Fig. 4c and Extended Data Fig. 7).
Specifically, the growth rates of humanized *E. coli* aligned with the residual activities of
recombinant ASL variants: D87G, which results in a complete loss of activity, could not
complement the arginine auxotrophy, while the least severe R12Q variant supported growth
to 84% of WT levels.

The ASL example demonstrates LEICA's versatility in screening enzymes from diverse
metabolic pathways. Its expandability suggests LEICA could be used as a platform to study a
wide range of human metabolic enzymes, further broadening the potential applications in
identifying pathogenic sequence variations and therapeutic compounds for metabolic
disorders.

DISCUSSION

High-throughput sequencing has revolutionized our understanding of the human genome,
providing vast amounts of data on genetic variations. The mere knowledge of genetic
sequences is insufficient, however; the key challenge lies in deciphering the functional
significance of these variations, particularly those associated with genetic disorders. Genome-
wide association studies (GWAS) have identified mutations linked to various disorders, but
understanding the causality of these mutations remains a significant hurdle.

To address this challenge, we developed a live bacteria assay, named LEICA, to measure
activities of heterologously expressed human enzymes. Unlike traditional *in vitro* assays and
cell-based assays, this system leverages the metabolic capabilities of bacterial cells to rapidly
characterize genetic variations associated with enzyme deficiencies. By coupling bacterial
growth with enzyme activity, this approach provides a simpler and more cost-effective
alternative to existing screening methods as it highly correlates with the activities reported
from previous studies. A notable feature of LEICA is its ability to screen for pathogenic
variations through the coupling of bacterial growth with enzyme activity. This alternative
approach offers distinct advantages over traditional screening methods, by rapidly providing
insights into the functional consequences of genetic variations.

Beyond its utility in studying genetic variations, LEICA proves effective for screening drug
effects on human enzymes. By directly assessing the impact of various compounds on
enzyme activity in an intracellular environment, it bridges the gap between *in vitro* assays and

cell-based assays. While offering advantages such as simplicity, speed, and cost-
effectiveness, it also presents **some** limitations, **such as** its inability to screen certain
compounds with antimicrobial properties. **However, incorporating a pairwise comparison of**
**the humanized *E. coli* with the strain carrying the endogenous bacterial gene effectively**
**prevents false-positive calls, as demonstrated with 6AN treatment (Fig. 2b, 3b, and Extended**
**Data Fig. 5). Also, we acknowledge that bacterial cells may have different permeability to**
**chemical compounds compared to human cells, which could affect the assay's representation**
**of drug efficacy in human cells. However, *E. coli* cells are non-selectively permeable to small**
**molecules (<600 Da)⁵³, and LEICA's demonstrated robustness in identifying seven effective**
**compounds from the drug library supports its reliability as a screening tool. Nonetheless, its**
**ability to operate under intracellular conditions makes it a valuable approach for compound**
**screening. Looking ahead, LEICA holds significant promise for high-throughput screening and**
**personalized drug testing (Supplementary Note 3).**

[revised manuscript text omitted]

The human *G6PD* gene was chemically synthesized (IDT, sequence available in
**Supplementary Note 4**) and cloned into the pTrcHis2A plasmid (Invitrogen) using Gibson
assembly (NEBuilder HiFi DNA Assembly Kit, NEB). Briefly, 5 fmol pTrcHis2A plasmid
backbone (linearized by PCR) and 20 fmol G6PD gene fragment were mixed in 6 μ l reaction
followed by incubation at 50°C for 15 min. Different G6PD variants were constructed by
assembling split human G6PD fragments (upstream and downstream of the variation)
amplified by primers containing genetic variations. Full G6PD constructs were assembled by
OE PCR as follows: 5 ng each of the fragments were ligated by 15 cycles of the following PCR
reaction: 98°C for 30 s, 68°C for 30 s, and 72°C for 90 s. Ligated products were amplified in
the same tube by adding two outermost primers (HsaG6PD_F and HsaG6PD_R) with 20
cycles of PCR reaction: 98°C for 30 s, 67°C for 30 s, and 72°C for 90 s. The assembled
constructs were cloned into pTrcHis2A using the aforementioned method. **Then, plasmid**
**carrying each *G6PD* variant was introduced into the *E. coli* 20.71 Δ *pgi* Δ *zwf* double knockout**
**strain to construct humanized G6PD *E. coli* model. Glycolytic flux was solely supported by the**
**HMP shunt in *E. coli* 20.71 Δ *pgi* strain or 20.71 Δ *pgi* Δ *zwf* having human *G6PD*, which induces**
**a slow growth rate, as reported elsewhere⁵⁵. Thus, we monitored the growth of these strains**
**for up to 60 hrs.** Primer sequences are summarized in **Supplementary Table 4**.

**Measuring effects of small molecules using LEICA**

The G6PD activator AG1, RRx-001, brimonidine (BMN), 6-aminonicotinamide (6AN),
proparacaine hydrochloride (PPC) were purchased from MedChemExpress (USA).
Tetracycline hydrochloride, metamizole (dipyrone) monohydrate (MMZ), and G6PDi-1, were
purchased from Sigma-Aldrich (USA). Dehydroepiandrosterone (DHEA) was bought from
ApexBio (USA). RRx-001 was prepared as a 100 mM dimethyl sulfoxide (DMSO) solution.
AG1, G6PDi-1, and DHEA were prepared as 30 mM DMSO solutions. BMN, PPC, MMZ, and
6AN were prepared as 10 mM aqueous solutions. Tetracycline was prepared as a 10 mg/ml
aqueous solution. Compounds were treated at appropriate concentrations with a final DMSO
concentration of 1%. DMSO was treated (1%) as an untreated vehicle control.

**Drug library screening**

**Cellular Metabolism Screening Library (Cayman Chemical, Cat #33705, Batch #0609421) was**
**used for drug library screening. 1.2 μ l of 10 mM compound solutions (DMSO) were dispensed**
**to 96-well microplates. *E. coli* 20.71 Δ *pgi* and *E. coli* 20.71 Δ *pgi* Δ *zwf* carrying pTrc_G6PD-**

WT was incubated in 3 ml LB medium containing 50 µg/ml kanamycin, 25 µg/ml
chloramphenicol, and 1 mM IPTG. 100 µg/ml carbenicillin was added to cells carrying plasmid.
1 ml of overnight cultures were washed with 1 ml of M9 glucose medium to remove excess
nutrients of LB. The washed resuspensions were inoculated into 3 ml fresh M9 glucose
medium (with initial OD of 0.03) containing 50 µg/ml kanamycin, 25 µg/ml chloramphenicol,
and 1 mM IPTG. 100 µg/ml carbenicillin was added to cells carrying plasmid. Then cultures
were incubated at 37°C for 3 hrs and 120 µl of cultures were transferred to 96-well microplates
having compounds resulting in 100 µM final concentration. Each plate comprises three
replicated cultures of 15 compounds and vehicle control (1% DMSO) for both *E. coli* 20.71
Δpgi and *E. coli* 20.71 $\Delta pgi \Delta zwf$ carrying pTrc_G6PD-WT. Cell growth was monitored every
60 min in a BioTek LogPhase 600 microplate reader (Agilent; set at 37°C with 800 rpm
shaking).

**Cloning ASL variants and construction of humanized ASL *E. coli* model**

The human ASL ORF clone was obtained from GenScript (Clone ID OHu22033, Acc.
NM_001024943.2) and cloned into the pTrcHis2A plasmid using Gibson assembly. Briefly, 5
474 fmol pTrcHis2A plasmid backbone (linearized by PCR) and 20 fmol ASL gene fragments were
475 mixed in 6 µl reaction followed by incubation at 50°C for 15 min. Different ASL variants were
476 constructed by assembling split human ASL fragments (upstream and downstream of the
477 variation) amplified by primers containing genetic variations. Full ASL constructs were
478 assembled by OE PCR as follows: 5 ng each of the fragments were ligated by 15 cycles of
479 the following PCR reaction: 98°C for 30 s, 68°C for 30 s, and 72°C for 90 s. Ligated products
were amplified in the same tube by adding two outermost primers (HsaASL_F and HsaASL_R)
with 20 cycles of PCR reaction: 98°C for 30 s, 67°C for 30 s, and 72°C for 90 s. The assembled
constructs were cloned into pTrcHis2A using the aforementioned method. Primer sequences
are summarized in Supplementary Table 4.

**Statistical analysis**

To compare the difference of means, a two-sided Welch's *t*-test was used with Bonferroni
correction for multiple hypothesis testing. Dose-response curve was fitted to the Hill curve to
estimate EC₅₀ or IC₅₀. For drug library screening, 15 compounds were first compared with
untreated control using a two-sided Welch's *t*-test corrected for multiple hypothesis testing
using Bonferroni method. Compounds that induced a significant growth difference (*p*-value <
0.001) in *E. coli* 20.71 $\Delta pgi \Delta zwf$ carrying pTrc_G6PD-WT strain, while not inducing a
significant difference (*p*-value ≥ 0.001) for *E. coli* 20.71 Δpgi were selected as primary hits.

**DATA AVAILABILITY**

The main data supporting the results in this study are available within the paper and its
Supplementary Information. Source data are provided with this paper. All compounds in the
drug library (n=160) are anonymized using proxy identifiers. For transparency, the actual
chemical names of all compounds have been provided to the journal's editorial team in a
separate document. Researchers interested in reproducing or building on this work may
contact the corresponding author to request access to the compound names and structures,
which will be made available under a confidentiality agreement to support ongoing research
and development efforts. Full disclosure of the identity of the candidate lead compounds is
deferred to a later date.

**ACKNOWLEDGEMENTS**

This work was funded by the Y.C. Fung Endowed Chair in Bioengineering at UC San Diego.

**AUTHOR CONTRIBUTIONS**

B. O. P. conceived and supervised the study. D. C. and B. O. P. designed the experiments.
D. C. performed the experiments. D. C. and B. O. P. analyzed the data and wrote the
manuscript. All authors read and approved the final manuscript.

**COMPETING INTERESTS**

All authors declare no competing interests.

[revised manuscript text omitted]
(LEICA). Growth rate of the *E. coli* 20.71 $\Delta pgi \Delta zwf$ strain carrying human GPI (humanized *E.*
*coli* for GPI) reflects activity of the human enzyme. EMPP: the Embden-Meyerhof-Parnas
pathway. HMP shunt: hexose monophosphate shunt. G6P: glucose-6-phosphate. 6PG: 6-
phosphogluconate. F6P: fructose-6-phosphate. FBP: fructose-1,6-bisphosphate. TCA cycle:
tricarboxylic acid cycle. **c,** Activities of human GPI variants were represented by growth rates
of the humanized *E. coli* for GPI. Data are presented as mean values \pm SD. Error bars
indicate SD of ten replicated cultures. Numbers are relative differences of growth rates
compared to the wild-type (WT) enzyme; in percentage. ns, no significant difference. **p*-value
= 0.022. ***p*-value < 0.005 (two-sided Welch's *t*-test with Bonferroni correction). Circles
indicate ten independent cultures. **d,** Growth rates of the *E. coli* with human gene swap shows
high linear correlation with previously reported activities of recombinant enzymes (Pearson's
R^2 of 0.96; dashed line)¹⁴. Data are presented as mean values \pm SD. Error bars indicate SD
of biological replicates (n=10). Specific activities of recombinant enzymes were pulled from a
previous report¹⁴. **e,** Growth of the double knockout strain expressing human G6PD (*E. coli*
$\Delta pgi \Delta zwf + pTrc_human$ G6PD; humanized *E. coli* for G6PD) is governed by activity of the
human G6PD. **f,** Growth rates of humanized *E. coli* for G6PD and catalytic constants (k_{cat}) of
recombinant enzymes²⁵⁻³⁰ had high correlation (Pearson's R^2 of 0.84; dashed linear
regression line). **Data are presented as mean values \pm SD. Error bars indicate SD of five**
**replicated cultures. WT: wild-type.** For G6PD-Volendam, the specific activity was used³¹,
because k_{cat} is not available. **g,** Activities of a few less-characterized human G6PD variants
were represented by growth rates of the humanized *E. coli*. Data are presented as mean
values \pm SD. Error bars indicate SD of five replicated cultures. Circles indicate five
independent data points.

**Figure 2. Effect of small molecules on glucose 6-phosphate dehydrogenase. a,** Effect of
a G6PD activator AG1 on growth rate of humanized *E. coli*. Half-maximal effective
concentration (EC_{50}) is calculated from the dose-response curve (dashed lines). Circles are
individual data points (n=10). **b,** Growth rates of humanized *E. coli* in LEICA with different
compounds. Vehicle: untreated control (1% DMSO). Tc: tetracycline. PPC: proparacaine.
6AN: 6-aminonicotinamide. RRx-001: bromonitroimidazole. BMN: brimonidine. DHEA:

dehydroepiandrosterone. G6PDi-1: inhibitor of G6PD 1. MMZ: metamizol. All compounds
were treated with the final concentration of 100 μ M, except for tetracycline (10 μ g/ml). ND: no
observable growth was detected. Data are presented as mean values \pm SD. Error bars show
SD of three replicates. Dots are individual data points. **ns: no significant difference. * p -value =**
**0.025 (BMN) and 0.0135 (G6PDi-1)** (two-sided Welch's t -test with Bonferroni correction,
compared to the untreated control). **c-e**, Dose-response validation of inhibitory effect of three
primary hits G6PDi-1 (**c**), DHEA (**d**), and brimonidine (**e**). Half-maximal inhibitory concentration
(IC_{50}) was calculated from a Hill curve (four parameters; dashed line) fitted to the dose
response curve. Dots are individual data points ($n=5$). * p -value < 0.05. ** p -value < 0.005. *** p -
value < 0.001 (two-sided Welch's t -test with Bonferroni correction, compared to the untreated
control).

**Figure 3. Drug library screening using LEICA. a**, Various compounds are supplemented to
LEICA and growth is monitored using a plate reader. Change in G6PD activity in response to
chemical compounds is reported as the growth difference of the humanized *E. coli* at a fixed
concentration compared to untreated control (primary hits). Primary hits are further validated
by dose-response assay, resulting in discovery of lead compounds. **b**, Growth rates of *E. coli*
expressing human G6PD compared to that of *E. coli* expressing endogenous G6PD. Seven
compounds induced significant (p -value < 0.001; two-sided Welch's t -test, compared to the
untreated control) growth difference, while inducing no significant difference on *E. coli*
expressing endogenous G6PD. All compounds were treated with the final concentration of
100 μ M. Data are presented as mean values \pm SD. Error bars show SD of three replicates.
**c**, Effect of primary hit compounds at different concentrations on humanized *E. coli* for G6PD
WT. Half-maximal effective concentration (IC_{50}) is calculated from the dose-response curve
(dashed lines). Circles are individual data points ($n=5$).

**Figure 4. Expanding LEICA for screening human argininosuccinate lyase (ASL). a**,
Schematic representation of the human urea cycle. BC: bicarbonate. CP: carbamoyl
phosphate. CIT: citrulline. ASA: argininosuccinate. ORN: ornithine. Fum: fumarate. **b**, Arginine
auxotrophy induced by *argH* knockout is complemented by human argininosuccinate lyase
(ASL). **c**, Growth rates of wild-type (WT) *E. coli*, *argH* knockout strain, and *argH* knockout
strain expressing WT and variants of human ASL. *in vitro* activities of the recombinant
enzymes^{12,52} are indicated above. ND: no observable growth was detected. Data are
presented as mean values \pm SD. Error bars show SD of five replicates. Dots are individual

data points. *** p -value < 0.001 (two-sided Welch's t -test with Bonferroni correction, compared
to the WT).

**EXTENDED DATA FIGURES**

**Extended Data Figure 1. Growth profiles of *E. coli* carrying human glucose-6-phosphate**
**isomerase (GPI) or its mutants in M9 glucose medium.** WT: wild-type human GPI. Data
are presented as mean values +/- SD. Error bars indicate SD of ten replicated cultures.

**Extended Data Figure 2. Growth of *E. coli* lacking phosphoglucose isomerase (*pgi*) or**
**glucose-6-phosphate dehydrogenase (*zwf*) in M9 glucose medium. a,** Growth profiles of
*E. coli* Δpgi and $\Delta pgi \Delta zwf$ strains in M9 glucose medium. Human glucose-6-phosphate
dehydrogenase (G6PD) rescued glucose utilization of $\Delta pgi \Delta zwf$ knockout strain. Data are
presented as mean values +/- SD. Error bars show SD of five replicates. **b,** Growth rates of
Δpgi and $\Delta pgi \Delta zwf$ strains. *E. coli* utilizing glucose with its endogenous G6PD (Zwf; *E. coli*
Δpgi) or human G6PD exhibited no significant difference in growth rate. Data are presented
as mean values +/- SD. Error bars show SD of five replicates. ND, no growth was detected.
758 ns, no significant statistical difference (p -value = 0.346, two-tailed Welch's t -test). Dots
represent individual data points.

**Extended Data Figure 3. LEICA monitors activities of G6PD variants as growth rates are**
**contingent on G6PD activity. a,** Growth profiles of humanized *E. coli* for G6PD expressing
well-characterized variants. **b,** Growth profiles of humanized *E. coli* for G6PD expressing or
less-characterized variants. Data are presented as mean values +/- SD. Error bars show SD
of five replicates.

**Extended Data Figure 4. Effect of various compounds on humanized *E. coli* for G6PD.**
**Growth profiles of humanized *E. coli* for G6PD (*E. coli* 20.71 $\Delta pgi \Delta zwf$ + *HsaG6PD*)**
**strain under different drug treatments.** veh: untreated control (1% DMSO). Tc: tetracycline.
PPC: proparacaine. 6AN: 6-aminonicotinamide. RRx-001: bromonitroizidine. BMN:
brimonidine. DHEA: dehydroepiandrosterone. G6PDi-1: inhibitor of G6PD 1. MMZ: metamizol.
All compounds were treated with the final concentration of 100 μ M, except for tetracycline (10
μ g/ml). Data are presented as mean values +/- SD. Error bars show SD of three replicates.

**Extended Data Figure 5. Effect of various compounds on *E. coli*.** **a,** Growth profiles of *E.*
*coli* *pgi* knockout strain under different drug treatments. veh: untreated control (1% DMSO).
Tc: tetracycline. PPC: proparacaine. 6AN: 6-aminonicotinamide. RRx-001: bromonitroizidine.
BMN: brimonidine. DHEA: dehydroepiandrosterone. G6PDi-1: inhibitor of G6PD 1. MMZ:
metamizol. All compounds were treated with the final concentration of 100 μ M, except for
tetracycline (10 μ g/ml). Data are presented as mean values +/- SD. Error bars show SD of
three replicates. **b,** Growth rates of the strain treated with the treatments. Data are presented

as mean values \pm SD. Error bars show SD of three replicates. Dots represent individual data
points. No significant difference in growth rate was observed (two-tailed Welch's *t*-test with
Bonferroni correction).

**Extended Data Figure 6. Growth profiles of wild-type (WT) *E. coli*, *argH* knockout strain,**
**and *argH* knockout strain expressing WT and variants of human ASL.** Data are presented
as mean values \pm SD. Error bars show SD of five replicates.

Point by point response to reviewers

Reviewer #1

Summary: The impact of the work has been significantly enhanced through the extensive inclusion of the following findings:

- Identification of seven potential lead compounds for G6PD inhibition within a 160-member drug library, with four of the hits being novel discoveries.
- Analysis of four less-characterized G6PD variants with no previously reported biochemical properties.
- Demonstration of a linear correlation between the growth rates of humanized *E. coli* expressing G6PD and catalytic constants (*k_{cat}*) for 13 additional well-characterized variants.
- Expansion of LEICA beyond glycolysis to screen the activity of human argininosuccinate lyase (ASL).

The revised manuscript now more effectively highlights the translational potential of this straightforward and cost-effective method for screening genetic variants within a real clinical patient pool.

The analytical limitations of the method are now more thoroughly addressed, with a detailed comparison to other gold-standard methods. The authors have clarified the specific application of LEICA for investigating enzyme activity independently of other pathogenic factors, emphasizing both its advantages and drawbacks.

All inquiries were carefully addressed and clarified, and all stylistic suggestions were implemented.

Optional comment: Regarding the drug library screening, did you encounter any false-negative results? Specifically, were there compounds known to inhibit G6PD but not identified as such by LEICA?

Response: We appreciate the reviewer's efforts and constructive suggestions in the review process, which helped improve the manuscript.

Regarding the question on false-negatives: Among the 160 compounds tested, we did encounter any compounds previously reported to inhibit human G6PD that failed to be identified by LEICA.

Reviewer #2

All of my concerns have been thoroughly addressed, and I now recommend acceptance.

Response: We sincerely appreciate the reviewer's feedback throughout the review process.

We are glad that our revisions have addressed your concerns, and we thank you for your recommendation for acceptance.

Reviewer #3

Since the initial submission, the authors have performed additional experiments and have made significant additions to the manuscript. In addition to expanding the work related to glycolytic enzymes, argininosuccinate lyase (ASL), an essential urea cycle enzyme, was also studied in the LEICA system. Interestingly, humanized *E. coli* expressing pathogenic ASL variants showed markedly reduced growth

rates compared to wild-type cells. I appreciate this addition, as the LEICA method has now shown applicability to metabolic pathways related to amino acid metabolism. As there are numerous biochemical genetic disorders related to impaired to the metabolism of amino acids (and of course other substrates), I look forward to following the results of this screening assay as it relates to an expanding number of conditions.

Response: Thank you for your encouraging and insightful feedback. We are glad that the additional experiments strengthened the manuscript. Your recognition of LEICA's potential to be applied to a wider range of metabolic pathways and genetic disorders is greatly appreciated. We share your enthusiasm for expanding this platform and look forward to future developments in this area.